



**A Skewed perspective of the Indian rainfall-ENSO Relationship**
**Justin Schulte[1*], Fredrick Policielli[2], and Benjamin Zaitchik[3]**
**1. Science Systems and Applications, Inc.**
**2. NASA Goddard Space Flight Center**
**3. John Hopkins University**
***corresponding Author: Justin Schulte (justin.a.schulte@nasa.gov)**
**Abstract**
The application of higher-order wavelet analysis to India rainfall and the El Niño/Southern Oscillation (ENSO) is
presented. An auto-bicoherence analysis is used to extract the frequency modes contributing to the skewness of India
rainfall and ENSO. A nonlinear wavelet coherence method is proposed for diagnosing why the time-domain
correlation between two time series temporally changes when at least one time series has changing nonlinear
characteristics.

13          The results indicate the India rainfall and ENSO are highly nonlinear phenomenon. It is also demonstrated
that the sea surface temperature (SST) patterns associated with different nonlinear ENSO modes depend on the
frequency components participating in the nonlinear phase coupling. The SST pattern associated with coupling
between ENSO modes with periods of 31 and 15.5 months is reminiscent of a central Pacific El Niño and intensifies
around 1995, contrasting with the coupling between the 62- and 31- month modes that became active around the 1970s
ENSO regime shift. A nonlinear coherence analysis showed that the skewness of India rainfall is weakly correlated
with that of 4 ENSO time series after the 1970s, indicating that increases in ENSO skewness after 1970's at least
partially contributed to the weakening India rainfall-ENSO relationship in recent decades. The implication of this
result is that the intensity of skewed El Niño events is likely to overestimate India drought severity, which was the
case in the 1997 monsoon season, a time point when the nonlinear wavelet coherence between All-India rainfall and
ENSO reached its lowest value in the 1871-2016 period.
**1. Introduction**

Precipitation variability across India is largely related to the seasonal Southwest and Northeast monsoon
systems involving changes in the prevailing low-level wind direction. Understanding the precipitation variability
across India is complex because India rainfall is a non-stationary, non-linear phenomenon that is influenced by
numerous large-scale climate patterns such as the El Niño/Southern Oscillation (ENSO; Walker and Bliss, 1932) and
the Indian Ocean Dipole (IOD; Ashok et al., 2001; Ashok et al., 2004) pattern. Predicting India rainfall has important
implications for the agriculture, human health, and economy of India, making the Indian monsoon an active area of
research despite early work on monsoon prediction extending back to the 1800s (Blanford, 1884).

An important source of predictability for the Indian monsoon is ENSO. During EL Niño years, droughts are
favored, while rainfall surpluses are favored during La Niña years. However, there is no one-to-one relationship
between ENSO and Indian rainfall.  As a result, summer rainfall predictions based on ENSO have proven challenging.
For example, the 1997/1998 EL Niño event was extremely strong yet climatological Indian monsoon conditions were
observed (Shen and Kimoto, 1999; Slingo and Annamalai, 2000). It is therefore important to understand why certain
El Niño events are not accompanied by monsoon failures.

There are a few reasons for the challenges faced when predicting Indian rainfall using ENSO. The first reason
is that the relationship between ENSO and India rainfall is non-stationary. As shown by Torrence and Webster (1999),
the relationship between ENSO and India rainfall cycles between periods of high and low coherence. Kumar et al.
(1999) found that the relationship between India rainfall and ENSO weakened in the 1970s and hypothesized that a
southward shift in Walker circulation anomalies associated with ENSO events and increased Eurasian spring and
winter surface temperatures was responsible for the weakening relationship. Other work suggests that the changing





ENSO-India rainfall relationship was the result of tropical Atlantic sea surface temperatures (SSTs) and the Atlantic
Multi-decadal Oscillation modulating the relationship (Lu et al., 2006; Kucharski et al. 2007; Kucharksi et al., 2009;
Chen et al., 2010). In contrast, Kumar et al (2006) and Fan et al. (2017) argued that the occurrence of different ENSO
flavors (Johnson, 2013) such as the Eastern Pacific and Central Pacific types could explain the changes in the ENSO-
India rainfall relationship. Other investigators adopted another perspective to explain changes in the ENSO-India
rainfall relationship and concluded that temporal undulations in the ENSO-India rainfall relationship are related to
statistical under sampling and stochastic fluctuations (Gershunov et al. 2001; van Oldenborgh and Burgers, 2005;
Delsole and Shukla, 2006; Cash et al., 2017). In a recent analysis, Yun and Timmermann (2018) showed that changes
in the ENSO-Indian rainfall relationship are consistent with a stochastically perturbed ENSO signal and argued that
changes in the ENSO-India monsoon relationship may not be related to external climate forcing mechanisms.
The second reason for the ENSO-related prediction challenges is that ENSO itself is a non-stationary
phenomenon. Using wavelet analysis, Kestin et al. (1998) found that the interannual variability of ENSO from 1930
to 1960 was dominated by a 4- to 7- year periodicity, whereas for the time period from 1960 to 1990, the interannual
variability was also dominated by a 2- to 5- year periodicity. A wavelet power spectral analysis conducted by Torrence
and Webster (1999) and Schulte (2016a) showed that ENSO signal energy in the 2- to 7-year period band undulates,
with the signal energy of the Niño 3.4 time series particularly pronounced after the 1960s (Schulte 2016a).
The nonlinear characteristics (e.g. skewness) of ENSO are also non-stationary and undergo interdecadal
changes (Wu and Hsieh, 2003). Numerous studies have reported an ENSO regime shift in the 1970s in which ENSO
began to evolve more nonlinearly than in previous decades (An, 2004; An and Jin 2004; An, 2009). It is a curious fact
that the ENSO regime shift of the 1970s coincided with the weakening ENSO-India rainfall relationship as
documented by Kumar et al. (1999). This observation begs the question as to whether nonlinear ENSO regime changes
are related to changes in the ENSO-India rainfall relationship.
Various mechanisms have been proposed for explaining ENSO skewness. Kang and Kug (2002) suggested
that the asymmetry between the magnitude of El Niño and La Niña events is related to the relative westward
displacement of zonal wind stress anomalies during La Niña events compared to El Niño events. Jin et al., (2003) and
An and Jin (2004) found that ENSO asymmetry is related to nonlinear dynamical heating (NDH), where the magnitude
of NDH is related to the propagation characteristics of ENSO. During strong El Niño events like the 1982/1982 and
1997/1998 events, SST anomalies were found to propagate eastward, with the eastward propagation tending to produce
more NDH compared to weak EL Niño events when NDH is minimal (An and Jin, 2004). Since the late 1970s there
has been a propensity for eastward propagation characteristics of ENSO (Santoso et al., 2013), contrasting with the
time period before the 1970s that consisted of the relatively weak El Niño events of 1957/1958 and 1972/1973 (An
and Jin, 2004; An, 2009). More recently, Su et al. (2010) showed that vertical temperature advection may have an
opposing effect on ENSO asymmetry and that the asymmetry in the extreme eastern equatorial Pacific is related to
meridional ocean temperature advection.
Previous investigators have used different metrics to quantify ENSO asymmetry. To measure the nonlinear
character of ENSO, An and Jin (2004) used time-domain metrics such as skewness and maximum potential intensity
(MPI) to quantify the skewness of SST anomalies and the skewness of individual ENSO events, respectively. An
(2004) applied a principal component analysis (PCA) to a 21- year moving window of tropical Pacific SST skewness
and found that the first PCA mode is characterized by positive skewness across the eastern equatorial Pacific and
negative skewness across the central equatorial Pacific. This pattern means that interdecadal changes in the
nonlinearity of ENSO is associated with positively skewed SST anomalies across the eastern equatorial Pacific,
implying that El Niño events are stronger than La Niña events. While the methods implemented in the aforementioned
studies provided important insights, they cannot reveal the frequency modes of ENSO that are contributing to the
skewness. Furthermore, the sliding-window approach is not local in the sense that it cannot quantify the strength of
nonlinearity at a point in time because skewness is calculated using a set of observations over some time interval.
While the MPI index does address the problem of quantifying the skewness of individual events, it also does not
provide any information regarding the frequency components contributing to ENSO skewness.
Recognizing the limitations of time-domain approaches, Timmermann (2003) conducted a bi-spectral
analysis of the Niño 3 anomaly time series, where a peak $(f_1, f_2)$ in the bi-spectrum means there is statistical phase



dependence among oscillators with frequencies $f_1$, $f_2$, and $f_1 + f_2$. That bi-spectral analysis revealed statistically
significant bi-spectral power at several frequency pairs, including (0.038, 0.038), (0.028, 0.028), (0.0225, 0.0225),
(0.0045, 0.032), and (0.0045, 0.045) [month$^{-1}$]. The peaks (0.0045, 0.032), and (0.0045, 0.045) [months$^{-1}$] were
identified with the nonlinear interactions among 18-year and 2-year variability. Although the analysis provided new
insights, the Fourier-based analysis could not reveal how the nonlinear nature of ENSO changed with time, an
important property to capture given how the nonlinear characteristics of ENSO are non-stationary (Santoso et al.,
2013). Much like the cross-wavelet power (Maraun and Kurths, 2004) and time-domain covariance, bi-spectral power
is not a bounded quantity and so high bi-spectral power does not always mean strong phase dependence.
In this study, the deficiencies associated with the above-mentioned techniques are addressed using higher-
order wavelet analysis, which allows for the quantification of frequency-dependent and non-stationary nonlinearities
in time series (Van Millagan, 2004, Elsayad, 2006; Schulte, 2016b). More specifically, the objectives of the paper are
the following: 1) quantify the nonlinearity of ENSO and Indian rainfall using higher-order wavelet analysis together
with recently developed statistical tests; (2) Determine if different nonlinear modes of ENSO are associated with
distinct SST patterns; and (3) develop nonlinear wavelet coherence methods to test the hypothesis that the breakdown
of the ENSO-India rainfall relationship in recent decades is related to the shift of ENSO from a linear regime to a
nonlinear one. The paper is organized as follows: In Section 2, data used are described. Section 3 includes the
description of the implemented methodologies. Results are presented in Section 4 and concluding remarks are
provided in Section 5.
**2. Data**
Monthly rainfall data for 5 homogenous regions (Parthasarathy et al. 1995a) were obtained from the Indian
Institute of Tropical Meteorology website (http://www.tropmet.res.in). The five homogenous regions called the
Peninsula, Northwest, Northeast, Central Northeast and West Central regions were constructed based on attributes
such as contribution to annual rainfall amount and regional/global circulation parameters (Parthasarathy et al. 1995a;
Azad et al., 2010). The variability of India rainfall was also analyzed using the all-India (Parthasarathy et al. 1995b)
rainfall time series, which is created by averaging representative rain gauges at various locations across India (Mooley
and Parthasarathy, 1984). The full monsoon season (June-September) and the late monsoon (August-September)
season were used to identify possible within-season variations in the relationships. All 6 rainfall time series considered
are continuous and span the time period from 1871 to 2016. To remove the influence of the annual cycle, the time
series was converted into anomaly time series by subtracting the 1871-2016 long-term mean for each month from the
individual monthly values. The anomaly time series were subsequently standardized by dividing them by their
respective 1871-2016 standard deviations. Because wavelet analysis focuses on specific frequency components that
are not impacted by long-term time-domain trends, no detrending of the data was performed.
The monthly data for the Niño 1+2, Niño 3, Niño 3.4, and Niño 4 indices (available at:
https://www.esrl.noaa.gov/psd/gcos_wgsp/Timeseries/Data/nino34.long.data) from 1871 to 2016 were used to
understand how the nonlinear characteristics of SSTs vary from one ENSO region to another. The Niño 1+2 index is
the average SST in the region with latitudinal boundaries 0° and 10°S and longitudinal boundaries 90°W and 80°W
and the Niño 3 index is the average SSTs in the region with latitudinal boundaries 5°N and 5°S and longitudinal
boundaries 150°W and 90°W. Variations in SSTs further west were described using the Niño 3.4 and Niño 4 indices,
where the Niño 3.4 index is defined as the average SST in the region bounded by 5°N and 5°S and 170°W and 120°W
and the Niño 4 index is defined as average SSTs in the region bounded by 5°N and 5°S and 160°E and 150°W. The
seasonal cycle was removed from these time series in the same way as it was removed from the rainfall time series.
Like the rainfall data, these data were not detrended.
The monthly SST data from 1871-2016 were based on the Hadley Centre Global Sea Ice and Sea
Surface Temperature (HadISST1; Rayner et al., 2003) The data at each grid point were converted to
monthly anomalies in the same way as they were computed for the ENSO and All-India time series.
**3. Methods**


### 3.1 Event Decomposition


To quantify the time-domain skewness of individual ENSO and India rainfall events, the ENSO and rainfall
time series were first decomposed into individual events using the event decomposition procedure outlined by Schulte
and Lee (2019). That is, a time series $x_1, x_2,...,x_N$ with data points located at the time points $t_1, t_2,...,t_N$ was partitioned
into subsequences comprising adjacent data points whose values are negative in the case of negative events and whose
values are positive in the case of positive events. A positive event was considered to begin at $t_i$ if $x_i > 0$ and $x_{i-1} < 0$.
The decay phase of a positive event beginning at $x_i$ was then defined as the time point $t_j$ such that $t_j \geq t_i$, $x_j > 0$, $x_{j+1} <$
$0$, and $x_k > 0$ for all $k$ such that $i \leq k \leq j$. Negative events were identified by switching the inequalities in the statements
above. After the event decompositions, the peak intensity of events was calculated, where the peak intensity of a
negative (positive) event was the minimum (maximum) value obtained by a data point within the event period $[t_i \ t_j]$.
The persistence of an event was defined as the number of points composing the event and the event intensity was
defined as
$$I = \sum_{i=1}^{M} y_i \tag{1}$$

where the $y_i$ are the $M$ data points composing the event. The duration and intensity of events were depicted using
event spectra (Schulte and Lee, 2019).

### 3.2 Wavelet Analysis


To better diagnose changes in time series statistics associated with India rainfall and ENSO, we adopted a
wavelet analysis. The continuous wavelet transform of a time series $X = \{x_n : n = 1,2,..N\}$ is given by
$$W_n(s) = \sqrt{\frac{\delta t}{s}} \sum_{n'=1}^{N} x_{n'} \psi_0 \left[ (n' - n) \frac{\delta t}{s} \right] \tag{2}$$

where $s$ is wavelet scale, $\psi_0$ is an analyzing wavelet, $\delta t$ is a time step (1 month in this study), and $n$ is time. The
sample wavelet power spectrum $|W_n(s)|^2$ measures the energy content of a signal at time $n$ and scale $s$. The commonly
used Morlet wavelet with angular frequency $\omega = 6$ was used throughout this paper because it balances time and
frequency localization. The reader is referred to Torrence and Compo (1998) and Grinsted et al. (2004) for details
about wavelet analysis.
Wavelet coherence was used to quantify the linear relationship between two time series as a function of
frequency and time. Wavelet coherence between two time series $X$ and $Y$ is given by
$$R_n^2(s) = \frac{|Ss^{-1} W_n^{XY}(s)|^2}{S\left(s^{-1} |W_n^X(s)|^2\right) S\left(s^{-1} |W_n^Y(s)|^2\right)}, \tag{3}$$

where $S$ is a smoothing operator (Grinsted et al., 2004) and $W_n^{XY}(s)$ is the cross-wavelet power spectrum. A coherence
value of 1 indicates the strongest possible association between two variables at the scale $s$ and time $n$. Large values of
wavelet coherence correspond to time points and scales for which the relative phase difference between two time
series varies little over a time interval. That is, two time series are perfectly coherent at the scale $s$ if for some constant
$c$   $\phi_n^X(s) - \phi_n^Y(s) = c$, where $\phi_n^X(s)$ is the phase associated with $X$ and $\phi_n^Y(s)$ is the phase associated with $Y$. If the
relative phase relationship is sufficiently stable, then the wavelet coherence will emerge as statistically significant
(Section 3.4).
In the context of the Indian monsoon, strong coherence between rainfall and a climate pattern (e.g. ENSO)
at a scale $s$ indicates shared temporal characteristics between a climate pattern and rainfall. Because theory supports a
causal link between ENSO and monsoon variability, strong coherence means that ENSO modulates rainfall. That is,
when ENSO is in a warm phase at the scale $s$, negative rainfall anomalies are preferred; when ENSO is in a cool phase,
the preference is reversed. As a result, the rainfall time series will inherit the temporal characteristics of the climate
forcing time series at a scale $s$. If the climate forcing time series is strongly periodic, then the otherwise noisy rainfall
time series could become periodic as well.



### 3.3 Higher-order Wavelet Analysis

Although the wavelet power spectrum is useful for quantifying the signal energy at a scale $s$ and time $n$, it cannot determine if there is a nonlinear relationship among different frequency components. In fact, the power spectrum can only fully describe time series in frequency space in the case of linear systems in which the output is proportional to the input (King, 1998). For nonlinear systems, higher-order moments exist, and the frequency decomposition of higher-order moments such as skewness is necessary for a more complete description of the time series. Thus, higher-order wavelet methods were adopted to determine the frequency components contributing skewness without assuming stationarity like Fourier-based bicoherence analysis.

The type of nonlinearities that produce skewness are quadratic nonlinearities in which the scales $s_1$, $s_2$, and $s_3$ satisfy the sum rule

$$\frac{1}{s_3} = \frac{1}{s_1} + \frac{1}{s_2} \tag{4}$$

and the wavelet phases satisfy

$$\phi_n(s_3) = \phi_n(s_1) + \phi_n(s_2). \tag{5}$$

These types of nonlinearities arise, for example, when a sinusoid is squared, in which case a harmonic is produced.

In this paper, the frequency components contributing to the skewness of a times series were quantified using local and global wavelet-based auto-bicoherence methods (Schulte, 2016b). Global auto-bicoherence was computed using the equation

$$bi_{global}^X(s_1, s_2) = \frac{\left|B_{global}^X(s_1 s_2)\right|^2}{\left(\sum_{n=1}^{N}|W_n^X(s_1)W_n^X(s_2)|^2\right)\left(\sum_{n=1}^{N}|W_n^X(s_3)|^2\right)}, \tag{6}$$

where

$$B_{global}^X(s_1, s_2) = \sum_{n=1}^{N}\widehat{W}_n^X(s_3)W_n^X(s_1)W_n^X(s_2) \tag{7}$$

is the global bi-spectrum and the hat denotes the complex conjugate. Identical to wavelet coherence, auto-bicoherence is bounded by 0 and 1, a value of 1 indicating the strongest possible phase coupling among the phases $\phi_n(s_3)$, $\phi_n(s_2)$, and $\phi_n(s_1)$ such that sum rule Eq. (5) is satisfied. A peak in the auto-coherence spectrum at $(s_1, s_2)$ means there is phase coupling between oscillatory modes with scales $s_1$, $s_2$, and $s_3$. High auto-bicoherence at $(s_1, s_2)$ can also mean that the same oscillatory modes are contributing to the skewness of the time series.

While the global auto-bicoherence spectrum is useful for identifying nonlinear triads, it cannot determine how the strength of phase coupling changes with time. To determine if the strength of the phase coupling changes temporally, the local auto-bicoherence spectrum (Schulte, 2016b) given by

$$bi_n^X(s_1, s_2) = \frac{\left|Ss_1^{-1}B_n^X(s_1,s_2)\right|^2}{S\left(s_1^{-1}|W_n^X(s_1)W_n^X(s_2)|^2\right)S\left(s_1^{-1}|W_n^X(s_3)|^2\right)}, \tag{8}$$

was computed, where $B_n^X(s_1, s_2)$ is the local bi-spectrum given as

$$B_n^X(s_1, s_2) = \widehat{W}_n^X(s_3)W_n^X(s_1)W_n^X(s_2). \tag{9}$$

In this study, we focused on the local diagonal slices of the auto-bicoherence spectrum, which consists of all points such that $s_1 = s_2$ so that Eq. (4) implies that $s_3 = s_1/2$. In this special case, the local auto-bicoherence spectrum was calculated using the equation

$$bi_n^X(s_1, s_1) = \frac{\left|Ss_1^{-1}B_{local}(s_1,s_1)\right|^2}{S\left(s_1^{-1}|W_n^X(s_1)W_n^X(s_1)|^2\right)S\left(s_1^{-1}\left|W_n^X\left(\frac{s_1}{2}\right)\right|^2\right)} \tag{10}$$





to reveal the time-evolution of auto-bicoherence estimates located along the diagonal slice in the global spectra. Bi-
phase corresponding to each point in the local auto-bicoherence spectrum was used to quantify the local cycle
geometry of the time series. Local bi-phase is given by

$$\psi_n(s_1, s_2) = \phi_n(s_1) + \phi_n(s_2) - \phi_n(s_3) \qquad (11)$$

and measures the skewness and asymmetries of waveforms. A bi-phase of 0° means that the relationship among the
scale components produces positive skewness with respect to a horizonal axis so that positive deviations from the
mean are larger than negative deviations from the mean. On the other hand, a bi-phase of 180° indicates negative
skewness with respect to the mean. Bi-phases near -90° or 90° indicate the presence of asymmetric cycle geometry
(King, 1998; Maccarone, 2014; Schulte, 2016b), indicating that a time series rises (falls) more quickly than it falls
(arises).
To be consistent with the wavelet power and coherence analyses, results for the higher-order wavelet analysis
were casted in terms of Fourier period rather than wavelet scale. The Fourier period corresponding to $s_i$ was denoted
by $p_i$, where the Fourier period is obtained by multiplying $s_i$ by 1.03 for the Morlet wavelet (Torrence and Compo,
1998). Thus, the local diagonal slice of the auto-bicoherence spectra were plotted using the Fourier period $p_1$
corresponding to $s_1$ as the vertical axis and time as the horizonal axis. High (or statistically significant) local auto-
bicoherence at $p_1$ and time $n$ means that there is phase dependence between modes with periods $p_1$ and $p_1/2$ at time
$n$ because $2p_3 = p_1$ according to Eq. (4) when $p_1 = p_2$. In other words, the local diagonal slice determines if there is
phase coupling between an oscillatory mode and its harmonic at various time points.

### 3.4 Statistical Hypothesis Testing

The statistical significance of all wavelet spectra was evaluated using the cumulative area-wise test (Schulte,
2016a; Schulte, 2018) to account for the simultaneous testing of multiple hypotheses (Maraun and Kurths, 2004;
Maraun et al., 2014). To perform the cumulative area-wise test, the point-wise test p-values associated with all points
in the wavelet domain had to be estimated using theoretical red-noise backgrounds for wavelet power and Monte Carlo
methods for wavelet coherence and auto-bicoherence (Torrence and Compo, 1998, Grinsted et al., 2004, Schulte
2016b). After the point-wise test implementations, the cumulative area-wise test was used to assess the statistical
significance of points in the wavelet domain by tracking how the normalized area of contiguous regions of point-wise
significance changed as the point-wise significance level was varied. The test was applied at the 5% cumulative area-
wise significance level using point-wise significance levels ranging from 0.02 to 0.98 because this choice of point-
wise significance levels was shown to result in the cumulative area-wise test outperforming the point-wise test in
terms of true positive detection for high signal-to-noise ratios despite how the cumulative area-wise test is more
stringent. Technical details of the testing procedure can be found in Schulte (2018) and in Appendix A.
To assess the statistical significance of the global auto-bicoherence estimates, a modified version of the
cumulative area-wise test was applied. In the modified version of the cumulative area-wise test, the normalized area
of patches was computed by dividing patch area by the product $\hat{s_1}\hat{s_2}$, where $\hat{s_1}$ is the mean first-coordinate of the patch
and $\hat{s_2}$ is the mean second coordinate. The means were calculated by assuming that the point-wise significance regions
are polygons with a set of vertices (Schulte et al., 2015). The reason for this modified normalized area is that dividing
area by say, $\hat{s_1}^2$, retained the correlation between normalized area and $s_2$. The test was applied using the same point-
wise significance levels that were used to assess the statistical significance of wavelet power and coherence.
To further assess statistical significance of wavelet quantities, a topological significance test (Schulte et al.
2015; Schulte 2018) and a cumulative arc-wise test was also applied to the wavelet spectra. The implementation of
the topological significance test involved the computation of the number of holes and contiguous point-wise
significance regions at a discrete set of point-wise significance levels, resulting in persistent homology profiles. The
topological significance and cumulative arc-wise tests were applied at the 5% significance level, and the point-wise
significance levels used ranged from 0.02 to 0.98. The critical levels of the test were estimated using Monte Carlo
methods by generating 1000 realizations of a red-noise process with lag-1 auto-correlation coefficients equal to that
of the input time series.





**3.5 Higher-order Coherence**

Although wavelet coherence spectra can provide information regarding how the relationship between two climate variables changes at a scale $s$, it cannot completely explain why the time-domain correlation between the climate variables temporally fluctuates. The reason is that linear wavelet coherence only examines how well the variance of one time series corresponds to the variance of another at a scale $s$ because linear coherence is determined by the wavelet power spectra of the time series. That is, linear coherence between two climate variables means that larger fluctuations in one time series produce larger fluctuations of another climate variable at the scale $s$. However, for two time series to be perfectly correlated in the time domain, higher skewness of one climate variable must also correspond to higher skewness of the other climate variable.

Recognizing that skewness is important for better understanding time-domain correlation changes, the quantity

$$Bi_n^2(s) = \frac{\left|Ss_{smooth}^{-1}B_n^{XY}(s_1,s_2)\right|^2}{S\left(s_{smooth}^{-1}\left|B_n^X(s_1,s_2)\right|^2\right)S\left(s_{smooth}^{-1}\left|B_n^Y(s_1,s_2)\right|^2\right)}, \tag{12}$$

called third-order coherence (nonlinear coherence, hereafter) was used to determine if changes in the skewness of $X$ are associated with changes in the skewness of $Y$ (see Appendix B for a more general definition). In Eq, (12), $s_{smooth}$ is one of the three scales, and $B_n^{XY}(s_1,s_2)$ is the third-order cross-wavelet power spectrum, which is the product of the bi-spectrum of $X$ and the conjugate of the bi-spectrum of $Y$, the higher-order analog of the cross-wavelet power spectrum. The word cross-bispectrum was not used to avoid confusion with cross-bicoherence analysis (Van Milligen,1995). Like wavelet coherence, the nonlinear coherence is bounded by 0 and 1, a value of 1 indicating that the bi-spectra of $X$ and $Y$ at $(s_1,s_2)$ are perfectly and linearly correlated. The statistical significance of nonlinear coherence was assessed using Monte Carlo methods and the cumulative area-wise test in the same way as it was used to assess the statistical significance of wavelet coherence.

Another way to interpret higher-order wavelet coherence is using linear and nonlinear modes. A linear mode $\gamma_{s_i}^X$ is the signal component of $X$ at the scale $s_i$ obtained by setting all wavelet coefficients to zero except those at $s_i$ and taking the inverse wavelet transform of the result. Because linear modes are only composed of a single frequency component, the local cross-correlation (coherence) between $\gamma_{s_i}^X$ and $\gamma_{s_i}^Y$ is only impacted by the variances of $X$ and $Y$ at $s_i$. On the other hand, nonlinear coherence measures the local cross-correlation between the skewness of $\gamma_{s_1}^X + \gamma_{s_2}^X + \gamma_{s_3}^X$ and $\gamma_{s_1}^Y + \gamma_{s_2}^Y + \gamma_{s_3}^Y$ or between $\gamma_{s_1}^X + \gamma_{s_{1/2}}^X$ and $\gamma_{s_1}^Y + \gamma_{s_{1/2}}^Y$ in the case that $s_1 = s_2$.

To better understand nonlinear coherence, we supposed that

$$\phi_n^X(s_1) - \phi_n^Y(s_1) = c_1 \tag{13}$$

$$\phi_n^X(s_2) - \phi_n^Y(s_2) = c_2 \tag{14}$$

$$\phi_n^X(s_3) - \phi_n^Y(s_3) = c_3 \tag{15}$$

for constants $c_1$, $c_2$, and $c_3$. Adding Eqs. (13) and (14) and subtracting Eq. (15) from the result produced the equality

$$\phi_n^X(s_1) + \phi_n^X(s_2) - \phi_n^X(s_3) - (\phi_n^Y(s_1) + \phi_n^Y(s_2) - \phi_n^Y(s_3)) =$$

$$\psi_n^X(s_1,s_2) - \psi_n^Y(s_1,s_2) = \psi_n^{bi}(s_1,s_2) = K, \tag{16}$$

for some constant $K = c_1 + c_2 - c_3$. Thus, if $X$ is perfectly nonlinear coherent with $Y$, then $X$ and $Y$ must be perfectly coherent at the three scales participating in the phase coupling. Even if the coherence is perfect at two scales, the relative bi-phase $\psi_n^{bi}(s_1,s_2)$ will fluctuate randomly if the relative phase difference at the remaining scale fluctuates randomly so that the nonlinear coherence will be low. Thus, if nonlinear coherence is high, then there must be some non-random relationship between $X$ and $Y$ at all three scales even if high linear coherence was not identified at one or more scales. This theoretical idea suggests that nonlinear coherence can uncover relationships that linear coherence cannot (see Figure S1 in supplementary material).



The relative bi-phase difference $\psi_n^{bi}(s_1, s_2)$ is the higher-order analog of the relative phase difference
between two time series. It measures how much the cycle geometry of one time series lags that of another. A lagged
bi-phase of 180° means that the skewness or asymmetry of the forcing time series is opposite to that of the response.
For example, if the forcing has positive skewness, then the response will have negative skewness. If the relative bi-
phase is 0°, then negative (positive) skewness of the forcing produces negative (positive) skewness of the response,
contributing to the positive time-domain correlation between the time series. Scales and time points for which
nonlinear coherence is high are where the relative bi-phase is stable.
Throughout this paper, we will focus on nonlinear coherence computed along the diagonal slices ($p_1 = p_2$)
of the time series bi-spectra. The nonlinear coherence spectra are then plotted using $p_1$ as the vertical axis and time as
the horizonal axis. High nonlinear coherence at $p_1$ and $n$ means that the skewness or asymmetry between $\gamma_{p_1}^X + \gamma_{p_1/2}^X$
and $\gamma_{p_1}^Y + \gamma_{p_1/2}^Y$ are locally cross-correlated.
To demonstrate the concept of nonlinear coherence, we considered a simple example in which the nonlinear
climate forcing time series was given by
$$F(t) = \cos(\frac{2\pi}{p_1} t + \varphi) + \gamma(t) \cos(\frac{2\pi}{p_3} t + 2\varphi) + W_F(t) \qquad (17)$$

and the response to the forcing was given as
$$R(t) = \cos(\frac{2\pi}{p_1} t + \varphi) + w_R(t), \qquad (18)$$

In Eq. (17), $\gamma(t)$ is a time-varying nonlinear coefficient, $w_F(t)$ is Gaussian white noise associated with the forcing,
$w_R(t)$ is Gaussian white noise associated with the response, $\varphi = 0$ is phase, and $p_1 = 2p_3 = 32$. The nonlinear
coefficient was assumed to be a linear function of time, i.e.,
$$\gamma(t) = t/500. \qquad (19)$$

The effect of the coefficient is to linearly increase the variance of $F(t)$ at $p_3 = 16$ and increase the strength of the
quadratic phase coupling between the modes with periods $p_3 = p_1/2 = 16$ and $p_1 = 32$.
As shown in Figure 1a, $F(t)$ (black curve) and $R(t)$ (thick green curve) evolve coherently from $t = 0$ to $t =$
200. After $t = 200$, $F(t)$ begins to noticeably exceed $R(t)$ at certain time points (e.g. $t = 430$) while the relationship
between them at other points is reversed (e.g. $t = 450$) in the sense that a positive forcing produces a negative response.
As a result, the correlation between $F(t)$ and $R(t)$ weakens (Figure 1b). An inspection of the wavelet coherence
spectrum (Figure 2a) reveals that the coherence at $p_1 = 32$ is strong and stable so that changes in the relationship
strength at that time scale is not the cause of the weakening time-domain correlation. The coherence at all other periods
is also stationary by construction so that it is not the changing relationship strength at any scale that is causing the
time-domain correlation weakening. However, the variance of $F(t)$ at $p_3 = 16$ increases with time (not shown) and
the coherence between $F(t)$ and $R(t)$ is also weak at that time scale, implying that larger fluctuations in $F(t)$ at $p_3 = 16$
are not accompanied by larger fluctuations in $R(t)$. Thus, variance increase of $F(t)$ is one reason for the weakening
time-domain correlation. However, both linear coherence and wavelet power cannot explain why the skewness of $F(t)$
increases, while the skewness of $R(t)$ is relatively stable (Figure 1c).
To further diagnose a cause of the weakening time-domain correlation, it is necessary to look at the auto-
bicoherence spectrum of $F(t)$ and the nonlinear wavelet coherence spectrum. An inspection of the local auto-
bicoherence spectrum of $F(t)$ (Figure 2b) reveals that the auto-bicoherence at $p_1 = 32$ is increasing with time, indicating
that the phase coupling between modes with periods $p_3 = 16$ and $p_1 = 32$ is strengthening with time. The bi-phase of
0°, as indicated by arrows pointing to the right, confirms that the phase coupling is contributing to the positive
skewness seen in Figure 1a to an increasing degree. Furthermore, the nonlinear coherence between $R(t)$ and $F(t)$ is
weak and mostly statistically insignificant at $p_3 = 32$ (Figure 2c), implying that the skewness of $F(t)$ produced from
the phase coupling between the modes $p_3 = 16$ and $p_1 = 32$ does not influence the skewness of $R(t)$. In other words,
the skewness of $\gamma_{16}^F + \gamma_{32}^F$ is uncorrelated with the skewness of $\gamma_{16}^R + \gamma_{32}^R$, where $\gamma_{16}^F + \gamma_{32}^F$ is the sum of the cosines
in Eq. (17) and the components of $W_F(t)$ at $p_3 = 16$ and $p_1 = 32$. The nonlinear mode $\gamma_{16}^R + \gamma_{32}^R$ is the sum of the



cosine in Eq. (18) and the components of $w_R(t)$ at $p_3 = 16$ and $p_1 = 32$. The weak nonlinear coherence also means that
$\psi_n^F(32,32) - \psi_n^R(32,32)$ fluctuates randomly (not shown). Thus, the skewness of $R(t)$ in the time-domain is
practically uncorrelated with the skewness of $F(t)$ because the skewness of $F(t)$ is solely related to the phase coupling
between the modes with periods $p_3 = 16$ and $p_1 = 32$ Thus, the increase in skewness of $F(t)$ also contributes to the
weakening time-domain correlation.

The lack of nonlinear coherence at time scales for which $F(t)$ is nonlinear has implications for empirical
prediction. At time points when $F(t)$ is positively skewed $R(t)$ is overestimated because $R(t)$ is not inheriting the
skewness of $F(t)$. In other words, a large forcing produces an unexpectedly small response. That is, if one created a
linear regression model based on the relationship between $F(t)$ and $R(t)$ from $t = 0$ to $t = 200$ one would find that a
forcing value of, say, 1 would produce a response close to 1. If the same model was used to predict $R(t)$ at, say, $t =$
430 one would predict that the forcing with value around 2 should result in a response near 2. However, because the
relatively large value $F(430)$ results from skewness and $R(t)$ is uncorrelated with its skewness, the response is only as
strong as the part of $F(t)$ not resulting from the quadratic phase coupling. The more nonlinear $F(t)$ becomes, the more
$F(t)$ will overestimate $R(t)$ when $F(t)$ is positively skewed. Similarly, the positive forcing produces a negative response
at $t = 450$ because of skewness and not simply a change in variance. Nonlinear coherence allows for the quantification
and identification of these time-domain aberrations.

The weakening relationship shown in Figure 1b could lead a researcher to believe that another direct forcing
must be directly influencing $R(t)$. This belief could lead to the applications of partial coherence (Ng, and Chan, 2012)
and partial correlation analyses to identify another influential forcing mechanism. However, in this case, there are no
other direct forcing mechanisms; the weakening time-domain relationship is solely related to how $F(t)$ transitioned
from a linear process to a nonlinear process. That is, the change is related entirely to how the skewness of $F(t)$ changed.
However, the phenomena influencing the linearity of $F(t)$ would be at least indirectly related to $R(t)$.
**4. Results**
**4.1 Event Decomposition of ENSO and Indian Monsoon time series**

The time series of the Niño 1+2 and Niño 4 indices together with the corresponding event spectra are shown
in Figures 3 and Figures 4, where we have chosen to show the results for the Niño 4 and Niño 1+2 indices because
they provide contrasting results. For the Niño 1+2 time series, a few recent notably intense (Figure 3b) warm events
are located around 1982/1983, 1997/1998, and 2015/2016 coinciding with the strongest El Niño events in recent
decades (McPhaden, 1999, Hu and Fedorov, 2017; Santoso et al, 2017). The event spectra for the Niño 3 and Niño
3.4 indices identified notably intense warm events that occurred after the 1970s (not shown). A few notably intense
events were also found in the late 1800s and early 1900s, indicating that intense ENSO events are not unique to recent
decades.

To visualize how skewness changes temporally, a 20-year sliding skewness analysis was conducted. As
shown in Figure 5a, the skewness of the Niño 1+2 index is enhanced during the early 1880s, near zero around the
1930's and early 1940's, and especially enhanced after the 1970s. It also appears that there is an upward trend in
skewness beginning around the 1940s, where the skewness peaks around 2000. In contrast to the Niño 1+2 index, the
skewness of the Niño 4 index becomes more negative after the 1970s, and the magnitude of the skewness is generally
smaller than that of the Niño 1+2 time series. This finding suggests that the transition of the Niño 1+2 time series to a
nonlinear regime was more pronounced than the transition associated with the Niño 4 time series. Interestingly, a 20-
year sliding skewness analysis of All-India rainfall reveals that the skewness of June-September All-India rainfall
remains close to zero until the 1990s despite the upward trend in Niño 1+2 skewness beginning in the 1940s (Figure
5a). However, the skewness of June-September All-India rainfall becomes more negative in the 1990s and 2000s, but
it is unclear if that negative skewness is related to ENSO because the skewness of the Niño 1+2 and Niño 4 indices
do not change as abruptly. Negative June-September All-India rainfall skewness is accompanied by enhanced positive
skewness of the Niño 1+2 indices prior to the 1940s, which is consistent with how All-India rainfall is negatively
correlated with the Niño 1+2 index time series during that time period (Figures 5b and 5c). Our results suggest that
All-India rainfall skewness is more correlated with ENSO skewness prior to the 1930s than it is in recent decades.



### 4.2 The time-domain Indian Rainfall-ENSO Relationship.


Given the non-stationaries in skewness can influence the time-domain correlation between two time series
(Figure 1b), it is reasonable to hypothesize that the All-India rainfall relationship with the Niño 1+2 and Niño 4 indices
could be non-stationary given that changes in Indian rainfall skewness do not always correspond with changes in
ENSO skewness. To test the hypothesis, a 20-year sliding correlation analysis was conducted between these ENSO
indices and All-India rainfall for the full (June-September) and late monsoon (August-September) seasons. The
correlation between the time series of the Niño 1+2 and Niño 4 indices and All-India rainfall was computed directly
without seasonal averaging.
As shown in Figure 5b, the relationship between full season India rainfall and the Niño 1+2 index generally
weakens from the 1800's to the 2000s. In contrast, the Niño 4 index relationship with All-India rainfall for the full
season appears to have no long-term trend, resulting in the Niño 4 index becoming more strongly correlated with All-
India rainfall than the Niño 1+2 index after the 1970's. The relationship between Indian rainfall and time series for
the Niño 3 and Niño 3.4 indices was also found to be relatively weak after the 1970s (not shown).
The stronger relationship between All-India rainfall and the Niño 4 index compared to the Niño 1+2
relationship with All-India rainfall after the 1970s is more evident in the late-season analysis (Figure 5c). An abrupt
weakening of the Niño 1+2-rainfall relationship occurs around the 1970's, with the relationship reversing around the
1990s. A comparison of Figures 5a and Figures 5c reveals that the weakening and reversal of the relationship occurs
during the time period when the Niño 1+2 index is especially skewed, suggesting that ENSO skewness changes could
be contributing to changes in the time-domain correlation between ENSO and All-India rainfall. However, we have
not shown that ENSO skewness exceeds a red-noise background (Sections 4.2 and 4.3) so that ENSO skewness
changes and time-domain correlation impacts could still be noise and unpredictable. Nevertheless, this reversal is
consistent with how Fan et al. (2017) found that the SST composite difference between drought and drought-free El
Niño years during the 1979-2012 period features warming across the central equatorial Pacific and cooling across the
eastern equatorial Pacific, whereas the SST composite for the 1978-1987 period features warming across the eastern
to central equatorial Pacific. It also noted that Niño 1+2 index-rainfall relationship is also relatively weak during the
late 1800's when Niño 1+2 skewness is relatively high (Figure 5a).
A rapid weakening is also seen in the Niño 4-rainfall relationship, but it appears to begin in the 1980's, which
is later than the Niño 1+2-rainfall relationship breakdown. Nevertheless, the weakening Niño 4-India rainfall
relationship coincides with the enhanced negative skewness of the Niño 4 index (Figures 5a). The fact that Niño 1+2
skewness is greater than Niño 4 skewness after 1970s and that the Niño 1+2 index relationship with All-India rainfall
weakens more abruptly than the Niño 4 index relationship with All-India rainfall suggests that skewness could at least
partially explain the temporal fluctuations in the relationships seen in Figure 5. Thus, a further investigation is needed
to better understand the temporal changes in ENSO statistics and their impact on the ENSO-India rainfall relationship.

### 4.3. Wavelet Power Analysis and Coherence


To better understand the non-stationarity of ENSO statistics, the wavelet power spectra associated with the
ENSO time series were computed (Figure 6). Enhanced variance in the 16- to 64-month band is seen after 1965 for
all the time series. For the Niño 3 and Niño 4 time series, there is also enhanced variance in the 16- to 64-month period
band from 1875 to 1895, whereas the enhanced variance persists to around 1905 for the Niño 3.4 time series. Another
important aspect of the wavelet power spectra is that the cumulative area-wise significance regions extend across
many periods. For example, in the wavelet power spectrum of the Niño 1+2 index, there is a period-elongated region
around 1997/1998 extending from a period close to 4 months to a period around 64 months. A similar feature is also
evident in the wavelet power spectrum of the Niño 3 and Niño 3.4 indices but appears to be less pronounced in the
wavelet power spectrum of the Niño 4 index. The appearance of holes in contoured regions suggests that there are
oscillatory modes with nearby frequencies (Schulte, et al., 2015), though the wavelet power spectra cannot determine
if there is phase coupling between the oscillatory modes.
The wavelet coherence spectrum shown in Figure 7, indicates that the All-India rainfall relationship with the
Niño 1+2 and Niño 4 indices in the 16- to 64-month period band breaks down after 1995, which is consistent with the


findings from the sliding correlation analysis shown in Figure 5. The relationship between rainfall and these ENSO
indices also weakens around 1925, but this weakening does not appear in the sliding correlation analysis. Note that
the lack of coherence after 1995 coincides with the enhanced ENSO variance, implying that higher ENSO variance
need not be associated with higher All-India rainfall variance at those time scales. This result implies that intense
ENSO events arising from variance in the 16- to 64-month period band need not correspond with unusual monsoon
seasons. Indeed, the 1997/1998 ENSO event, which coincides with high power in the 16- to 64-month period band,
was associated with a near-normal 1997 monsoon season. More generally, these results imply that the difference Niño
1+2 – AIR is periodic in the 16- to 64-month period band, where AIR is All-India rainfall. The periodic property was
confirmed by computing the wavelet power spectrum of Niño 1+2 – AIR (supplementary Figure S2), with the
periodicity implying that time periods when ENSO overpredicts and underpredicts rainfall occur in regular intervals.
Thus, the result contradicts previous findings suggesting that the relationship between ENSO and Indian rainfall
fluctuates randomly (Yun and Timmermann, 2018). In other words, changes in ENSO variance could be contributing
to the weakening time-domain correlation. However, ENSO skewness is also enhanced during this time period (Figure
5a) so that weakening relationships may not be simply related to ENSO variance.
Averaging wavelet coherence in the 16 to 64-month period band further illustrates how the wavelet coherence
varies temporally (Figure 8). For example, wavelet coherence with both the ENSO indices reaches approximately 0.8
around 1975 before falling below 0.1 in the mid 1990s.  Because the coherence plots shown in Figure 7 are similar, it
is difficult to diagnose why the sliding correlation curves shown in Figure 5 have different temporal structures. For
example, the period-averaged coherence shown in Figure 8 between rainfall and both the ENSO indices are identical
around 1998 yet the relationship between the Niño 1+2 and All-India rainfall is weaker than the relationship between
the Niño 4 index and All-India rainfall around that time (Figures 5b and 5c). Thus, a further analysis is needed to
extract information unrevealed by the linear wavelet power and coherence methods.
**4.2. Global Auto-bicoherence**
**4.2.1 ENSO**
As a first step for better understanding the All-India rainfall-ENSO correlation curves shown in Figure 5, the
global auto-bicoherence spectra associated with the ENSO time series were computed (Figure 9). For all four ENSO
metrics, statistically significant auto-bicoherence was identified, with the global auto-bicoherence spectrum of the
Niño 1+2 index containing the greatest number of statistically significant auto-bicoherence estimates. A few notable
peaks in the Niño 1+2 index auto-bicoherence spectrum are located at (148, 105), (148, 52), (62, 44), and (88, 88)
[months]. The auto-bicoherence peak at (88, 88) suggests that there is phase coupling between an 88-month mode (~
7 years) and a 44-month mode (~ 3.5 years). The auto-bicoherence spectrum of the Niño 3, Niño 3.4, and Niño 4
indices all contain statistically significant auto-bicoherence peaks at (31, 31), implying phase coupling between a 31-
month mode and a harmonic with a period of 15.5 months. For the Niño 3.4 index, there is also an on-diagonal peak
at (55.6, 55.6), whereas for the Niño 3 index the peak is slightly shifted and located at (62, 44). A third peak in the
Niño 3.4 spectrum was found at (105, 47), which could be associated with decadal-scale amplitude modulations of
ENSO, though the peak does not correspond to the linkage between the 18-year and 2-year variance identified by
Timmermann (2003). The differences among the auto-bicoherence spectra suggests that the nonlinear character of
SSTs varies spatially, which is consistent with prior work showing how skewness is generally highest in the eastern
equatorial Pacific and lowest in the central equatorial Pacific (An and Jin, 2004).
To confirm the spatial heterogeneity in the nonlinear characteristics of SSTs, the auto-bicoherence associated
with SSTs at a few select peaks ($p_1$, $p_2$) were computed at each grid point in the domain bounded by 20°N and 20°S
and by 146°E and 80°W. The peaks were selected based on the auto-bicoherence spectra of the Niño 3.4 and Niño
1+2 indices. To select the peaks, local maxima in auto-bicoherence within the statistically significance regions shown
in Figure 9 were identified.
The spatial structure of auto-bicoherence corresponding to the peaks in the Niño 3.4 auto-bicoherence
spectrum are shown in Figure 10. The auto-bicoherence associated with the pair (31, 31) is greatest across the central
equatorial Pacific, with the overall spatial pattern being reminiscent of a central Pacific El Niño (Lee and McPhaden,
2010). This result suggests that the phase coupling between the 31-month mode and the 15.5-month mode could be



related to the occurrence of central Pacific El Niño events (Section 5). In contrast, the auto-bicoherence pattern
associated with the pair (56, 56) is more uniform, with auto-bicoherence slightly greater across the extreme eastern
equatorial Pacific than the central equatorial Pacific. This pattern is reminiscent of an eastern Pacific El Niño. Like
the pattern corresponding to the pair (31, 31), the auto-bicoherence for the pair (105, 57) tends to be greater across the
central equatorial Pacific. Our findings suggest that different nonlinear modes contribute to different ENSO flavors.
Although An and Jin (2004) and Burgers and Stephenson (1999) showed that skewness is greatest across the eastern
equatorial Pacific, we determined that such a time-domain approach is unable to capture frequency-dependent patterns
in nonlinearity.
The spatial auto-bicoherence plots associated with the peaks in the Niño 1+2 auto-bicoherence spectrum are
shown in Figure 11. The auto-bicoherence associated with the pairs (148, 53) and (148, 105) is strong across the
eastern equatorial Pacific but weak across the central equatorial Pacific, suggesting that the phase coupling between
the 148- and 105-month modes and between the 148- and 53-month modes are associated with the skewness of eastern
equatorial Pacific SSTs. The pattern associated with the pair (62, 44) is reminiscent of an eastern Pacific El Niño and
the auto-bicoherence associated with the pair (88, 88) is relatively weak across the entire equatorial Pacific. A
comparison of Figures 10 and 11 shows that there is a tendency for auto-bicoherence to be greater across the eastern
equatorial Pacific than the central equatorial Pacific, which is consistent with the results of An and Jin (2004) and
Burgers and Stephenson (1999) who found that SSTs across eastern equatorial Pacific are most skewed. The results
are also in agreement with Figure 5a, which shows how the magnitude of Niño 1+2 skewness is greater than that of
the Niño 4 skewness after the 1970s.

### 4.2.2 India Rainfall

The global auto-bicoherence spectra for the rainfall time series are shown in Figure 12. For all the rainfall
time series except for the central Northeast time series, statistically significant auto-bicoherence was identified. The
auto-bicoherence spectrum of the All-India time series contains four on-diagonal peaks, one located around (4,4),
another located at (18, 18), and two more located around (40, 40) and (90, 90) [months]. Each of these peaks indicate
time series components with periods 4, 18, 40, 90 months are phase coupled to the corresponding harmonics with
periods of 2, 9, 20, and 45 months. Such phase coupling is inconsistent with the null hypothesis of red noise, which
agrees with the findings of Schulte (2019) who found robust evidence that there are features embedded in the India
rainfall time series that exceed a red-noise background. Thus, it is natural to ask if these peaks are inherited from a
nonlinear climate forcing. For example, the peak (90, 90) in the All-India rainfall auto-bicoherence spectrum
corresponds well with the peak found in the auto-bicoherence spectra of the Niño 1+2 time series (Figure 9).
Figure 12 also reveals how the nonlinear characteristics corresponding to each region differ. The statistically
significant auto-bicoherence for the Peninsula, Northwest, West Central, and Northeast time series is mainly located
in regions for which $p_1$ and $p_2$ are less than 16 months. However, a peak at (256, 32) was found in the auto-bicoherence
spectrum of the Northeast time series, suggesting that the time series components with periods 28, 32, and 256 are
phase dependent. Many other differences are also seen through an inspection of Figure 12. Our findings suggest that
the processes governing precipitation variability in each of the regions differ (Roy and Tedeschi, 2016).

### 4.3 Local auto-bicoherence

### 4.3.1 ENSO

To determine if the strength of the identified nonlinearities changes with time, the local diagonal slices
corresponding to the global auto-bicoherence spectra shown in Figure 9 were computed. The results shown in Figure
13 reveal that the auto-bicoherence spectra of all ENSO time series contain statistically significant local auto-
bicoherence, but the spectrum of the Niño 4 index is only associated with a few statistically significant regions such
as the one around 2015 at a period of 32 months.
For the Niño 3 and Niño 3.4 time series, two features of interest are seen in the time period extending from
1973 to 2017 in the 16- to 64-month period band. The first feature is the time-elongated region of statistical
significance extending from 1973 to 2016 around a period of 61 months. This result implies that after 1973 the





nonlinear phase coupling between modes with periods of approximately 30.5 and 61 intensifies. This intensification
is consistent with studies showing that ENSO underwent a regime shift in the 1970s in which ENSO began to evolve
more nonlinearly than in previous decades (Santoso et al., 2013). This intensification is also evident in the Niño 1+2
auto-bicoherence spectrum, though the exact periods associated with the phase-coupled oscillatory modes are more
difficult to discern. Nevertheless, a comparison of Figures 5a and 13 reveals that enhanced skewness coincides with
stronger auto-bicoherence in the 32- to 64-month period, suggesting that the skewness partially arose from the stronger
phase coupling among modes with periods ranging from 32 to 64 months. The correspondence between auto-
bicoherence and time-domain skewness also holds for the Niño 3 and Niño 3.4 time series (not shown). Our findings
suggest that phase coupling among modes embedded in the 32- to 64-month period band plays an important role in
generating the skewness of ENSO warm events.

The second feature of interest in Niño 3 and Niño 3.4 auto-bicoherence spectra is the one that emerges around
1995 at a period of 31 months. Despite how recent studies indicate that the ENSO regime shift occurred around 1973,
this result suggests that the onset of this phase coupling occurred well after the 1970s regime shift just before the
1997/1998 El Niño event. Thus, the nonlinear character of, say, the 1982/1983 El Niño is different from that of both
the 1997/1998 and 2015/2016 El Niño events because of the additional phase coupling between the 15.5- and 31-
month modes. It is also noted that Figure 13 also shows that there are other time periods when ENSO behaved
nonlinearly, and so the recent nonlinear events may not be unique to recent decades. For example, the auto-bicoherence
spectrum of the Niño 3.4 time series is associated with enhanced auto-bicoherence around 1875 in the 32- to 128-
month period band. Nevertheless, our findings reveal that the stationarity of the phase coupling in recent decades is
unprecedented with respect to any other time period.

To confirm that the nonlinear phase coupling identified in Figure 13 is associated with skewed waveforms,
we inspected the corresponding local bi-phase spectra (not shown). It was found that the bi-phase in the 42- to 64-
month period band is generally  0° so that the nonlinear phase coupling in that period band contributes to the positive
skewness of the 1982/1983, 1997/1998, and 2015/2016 events.

The temporal change in the auto-bicoherence associated with the Niño 1+2 and Niño 4 indices was further
illustrated by averaging the local auto-bicoherence in the 32- to 64-month period band. As shown in Figure 8, the
auto-bicoherence associated with both ENSO indices increases after the 1970s. This increase in auto-bicoherence
coincides with the increase in skewness shown in Figure 5. Thus, the skewness of the Niño 1+2 and Niño 4 indices
appears to be related to the auto-bicoherence in the 32- to 64-month period band. It also noted that the auto-bicoherence
associated with the Niño 1+2 index peaks around 1998, which is consistent with how the 1997/1998 Niño 1+2 warm
event arose from nonlinear processes (An, and Jin, 2004). The auto-bicoherence was also high around the nonlinear
event 1982/1983 event (An and Jin, 2004), further supporting the idea that the skewness of individual Niño 1+2 warm
events is connected to the nonlinear phase coupling in the 32- to 64-month period band.

### 4.3.2 Local Bicoherence of India Rainfall and Non-linear Coherence

The local auto-bicoherence spectra of the India rainfall time series are shown in Figure 14. The statistically
significant auto-bicoherence was identified for all six time series, mainly for periods less than 64 months. The results
suggest that the phase coupling is many among higher frequency modes. However, for the All-India rainfall time
series, the auto-bicoherence spectrum reveals two time periods of statistically significant auto-bicoherence in the 64-
to 128-month period band. The first region extends from 1885 to 1925 and the second region extends from 1945 to
around 1985. The nonlinearities found in the India rainfall auto-bicoherence spectra were also found to be cumulative
arc-wise significant, though some differences in the results were found (Figure S3 in supplementary material). The
statistical significance of the results was further checked using the topological significance test (Schulte, 2019), which
also provided evidence that the time series are nonlinear (Figure S4 supplementary material).

To determine if the nonlinearities identified for All-India rainfall is related to ENSO, nonlinear coherence
was computed along the local diagonal slices of the auto-bicoherence spectra for both All-India rainfall and the four
ENSO metrics considered in this study. Furthermore, All-India rainfall is generally more strongly coherent with ENSO
than rainfall associated with the individual rainfall regions (Schulte, 2019) so only the results for All-India rainfall are
shown for brevity.





The results shown in Figure 15 indicate that the nonlinear wavelet coherence between All-India rainfall and
the time series for the all four ENSO indices is statistically significant in the 32- to 64-month period band. The
nonlinear coherence in this period band appears to peak around the 1972/1973 El Niño event, indicating that an
increase in positive skewness of ENSO should tend to coincide with enhanced negative skewness of All-India rainfall
around this time. As shown in Figure 8, the nonlinear coherence averaged in the 32- to 64-month period band fluctuates
less than linear coherence and reaches a clear global maximum around 1972/1973 before rapidly declining to a global
minimum around the 1997/1998 El Niño event when the Niño 1+2 index is very nonlinear (Figures 8 and 13).
Therefore, according to the discussion in Section 3.5, changes in ENSO skewness contributed to the weakening
relationships between ENSO and All-India rainfall shown in Figures 5a and 5b.
**5. Discussion/Conclusion**
The nonlinear nature of both ENSO and Indian rainfall were examined using higher-order wavelet methods.
The auto-bicoherence spectra of the four ENSO time series revealed that ENSO skewness arose from the phase
coupling of modes with various periods. The Niño 3.4 time series was found to contain coupling between modes with
period 31 and 15.5 in addition to coupling between modes with period of 61 months and 30.5 months. The phase-
coupling between the 31 and 15.5 modes was found to be especially strong after 1995, whereas the phase coupling
between the 61- and 30.5-month modes was found to intensify after the 1970s. The stronger phase coupling after the
1970s is consistent with how ENSO underwent a regime shift in the 1970s (Santoso et al., 2013), which was marked
by an increase in ENSO skewness.
The evolution of SSTs across the Niño 4, Niño 3.4, Niño 3, and Niño 1+2 regions was found to be nonlinear,
but the degree to which the time series are nonlinear are different. Overall, the Niño 1+2 time series was found to be
the most nonlinear, while the Niño 4 index was found to be the most linear. The spatial patterns associated with the
nonlinearities depend on the frequency components contributing to the nonlinearities. For example, phase coupling
between the modes with periods of 31 and 15.5 months was found to be strongest in the central equatorial Pacific and
weakest across the eastern equatorial Pacific. This finding suggest that the occurrence of central Pacific El Niño events
could be linked to this phase coupling, which is relevant to understanding the Indian monsoon because central Pacific
El Nino events have been shown to be more effective at creating drought-inducing subsidence over India (Kumar et
al., 2006).
The results from the present and previous studies (Fan et al. 2017) supports the idea that changes in the
ENSO-India rainfall relationship are related to ENSO flavors because ENSO nonlinearity appears to be related to
ENSO flavors (Figures 10 and 11), opposing the findings of other work showing that the changes are related to
sampling variability or to noise. According to Yun and Timmermann (2018), the changes in the time-domain
correlation between All-India summer rainfall (ISMR) and ENSO is consistent with the assumption that ISMR is the
sum of the ENSO signal and Gaussian white noise (i.e., ISMR = ENSO + white noise). However, for this hypothesis
to hold, the difference ISMR – ENSO must be Gaussian white noise. As shown in this study, the nonlinear wavelet
coherence between ENSO metrics and All-India rainfall is weak, which means that the difference ISMR – ENSO will
have non-Gaussian noise features so that ISMR is not consistent with a stochastically perturbed ENSO signal. The
retention of non-Gaussian noise features is certainly the case for $R(t) – F(t)$ in the example in Section 3.5 because the
difference would retain the cosine function with a period of 16. In the case of ISMR, the lack of nonlinear coherence
results in periodic behavior of ENSO – ISMR, which means that Indian rainfall is not simply a stochastically perturbed
ENSO signal, as noise does not contain periodicities. In contrast, if ISMR and ENSO were highly nonlinear wavelet
coherent, then they would have the same frequency components contributing to skewness and the difference of the
two would remove the skewness. Although our results cannot preclude noise as a contributor to fluctuations in the
time-domain correlation, the periodic nature of ENSO – ISMR does suggest that monsoon forecast error for a forecast
based on ENSO may be predictable to some extent.
The fact that nonlinear coherence between rainfall and ENSO is determined by linear coherence between
ENSO and rainfall at two or three frequencies means that the changing time-domain correlation could be more fully
understood by determining why linear coherence changes at the frequencies that contribute to ENSO skewness. Such
an analysis could provide a more mechanistic perspective than the theoretical perspective adopted in this study. A
preliminary analysis showed that enhanced linear coherence between the North Atlantic Oscillation index and All-


India rainfall after 1995 in the 16- to 64-month period band associated with ENSO nonlinearity. This result suggests
that conditions across the North Atlantic (Kakade, 2000, Bhatla, 2016) could influence the nonlinear coherence
between ENSO and All-India rainfall and thus the corresponding time-domain correlation.

The higher-order wavelet analysis conducted in this study also revealed that the nonlinear nature of the
rainfall time series for the regions considered varied. Our results are consistent with the findings from previous work
showing how the physical mechanisms governing precipitation variability are different (Roy and Tedisch, 2016).
However, the higher-order wavelet analysis conducted in this study allowed us to determine the time scales on which
the rainfall times series features differ. Further research is needed to fully understand why the nonlinear characteristics
differ from one region to another. Future work could include conducting nonlinear coherence analyses between indices
of various climate modes and the rainfall times series for each region individually.

A few other possible physical mechanisms behind the nonlinearity of the rainfall time series were examined.
For example, we computed the auto-bicoherence spectrum of the IOD and sunspot time series because they have been
postulated as climate drivers of Indian Rainfall (Ashok et al., 2001; Ashok et al., 2004; van Loon and Meehl, 2012).
Although these time series were found to be highly nonlinear, the auto-bicoherence spectra of them did not correspond
well with the rainfall time series. We found that the IOD contains strong coupling between the modes with periods of
256 and 128 months and between modes with periods of 128 months and 64 months (Figures S5 and S6 in
supplementary material), but no such coupling was found for any of the rainfall time series. Similarly, the sunspot
cycle time series was associated with strong coupling between 128 -and 256-month modes (Figures S7 and S8 in
supplementary material) but again no such coupling was identified in the rainfall time series. Future work could thus
include better understanding the physical mechanisms underlying the nonlinearities identified in this study.

The tools used and developed in this study may have important applications in understanding how forecasting
systems replicate Indian rainfall and its associated teleconnections. These methods, for example, could determine if
forecasting systems can reproduce nonlinear characteristics of climate time series. These identifications could provide
new directions for improving current forecasting systems and ultimately predictions of Indian rainfall.







**Appendix A**

The first step (STEP 1) in assessing the cumulative-area significance of a point was the calculation of the $N$
$= 12$ sets

$$P_{pw}^i = \left\{(b,a): \rho_{pw}(b,a) < \alpha_i\right\}, \tag{A1}$$

where each set is the subset of the wavelet domain consisting of points whose wavelet quantities are point-wise
statistically significant at the $\alpha_i$ significance level. In this paper, $\alpha_1 = 0.02$, $\alpha_{12} = 0.18$, and $\alpha_{i+1} - \alpha_i = 0.02$. In the
second step (STEP 2), a geometric pathway about $x$ was computed, where a geometric pathway is a nested sequence

$$P_1^x \subseteq P_2^x \subseteq \cdots \subseteq P_N^x \tag{A2}$$

such that the

$$P_i^x = \left\{(b,a): (b,a) \in P_{pw}^i, (b,a){\sim}x\right\} \tag{A3}$$

are path-components of $P_{pw}^i$ containing $x$. The equivalence relation $\sim$ on $P_{pw}^i$ makes two points in $P_{pw}^i$ equivalent if
they can be connected by a continuous path in $P_{pw}^i$. The third step (STEP 3) involved the calculation of the normalized
area corresponding to $P_i^x$. The normalized area is defined as patch area divided by the square of mean scale coordinate
of the patch, where $A_i^x$ was assumed to be 0 if $P_{pw}^i = \phi$ or $P_{pw}^i = \{x\}$. The critical area $A_i^{crit}$ was obtained by computing
the $(1 - \alpha_c)$th percentile of the null distribution of normalized areas corresponding to the significance level $\alpha_i$, where
$\alpha_c$ is the significance level of the cumulative area-wise test. The null distributions were constructed by generating
1000 patches at the $\alpha_i$ significance level under the null hypothesis of red noise. More specifically, realizations of a
red-noise process with lag-1 autocorrelation coefficients equal to that of input time series were used to create the
wavelet spectra from which the 1000 patches were obtained. The length of the realizations was set to 200, though the
length is irrelevant because patch area is not related to time series length but to the reproducing kernel of the analyzing
wavelet (Schulte 2019). The final step (Step 4) was to compute

$$r^x = \frac{1}{N}\sum_{j=1}^N \lambda_j^x, \tag{A4}$$

where $\lambda_j^x = 2$ if $P_j^x/A_j^{crit} > 1$, $\lambda_j^x = 0$ if $P_j^x/A_j^{crit} <= 1$, and $A_j^{crit}$ is the critical area associated with $\alpha_j$. The wavelet
quantity at the point $x$ was deemed statistically significant at the $\alpha_c$ cumulative area-wise level if $r^x > 1$.





**Appendix B**
For $p > 1$, the $(p+1)$-th order poly spectrum of a time series $X$ is given by
$$B_n^X(s_1, s_2, \dots, s_p) = \widehat{W}_n^X(s_{p+1})\left(\prod_{k=1}^{p} W_n^X(s_k)\right) \tag{B1}$$

where
$$\frac{1}{s_{p+1}} = \sum_{k=1}^{p} \frac{1}{s_k} \tag{B2}$$

The third-order poly spectrum is the bi-spectrum, and the fourth-order poly spectrum is the tri-spectrum (Collis et al.,
1998), which identifies the frequency components contributing to kurtosis. The $(p+1)$-th order coherence between two
time series is given as
$$R_n^2(s) = \frac{\left|S s_{smooth}^{-1} B_n^{XY}(s_1, s_2, \dots, s_p)\right|^2}{S\left(s_{smooth}^{-1}\left|B_n^X(s_1, s_2, \dots, s_p)\right|^2\right) S\left(s_{smooth}^{-1}\left|B_n^Y(s_1, s_2, \dots, s_p)\right|^2\right)}, \tag{B3}$$

where $B_n^{XY}(s_1, s_2, \dots, s_p)$ is the $(p+1)$-th-order cross-spectrum given by
$$B_n^{XY}(s_1, s_2, \dots, s_p) = B_n^X(s_1, s_2, \dots, s_p)\hat{B}_n^Y(s_1, s_2, \dots, s_p). \tag{B4}$$

When $p = 2$, Eq. (B3), measures the local cross-correlation between skewness, and when $p = 3$ the equation
measures the local cross-correlation between kurtosis.








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






Figure 1. (a) An idealized nonlinear forcing time series together with an idealized response $R(t)$. The 120-point sliding correlation between $F(t)$ and $R(t)$. (c) The 120-point sliding skewness of $F(t)$ and $R(t)$.



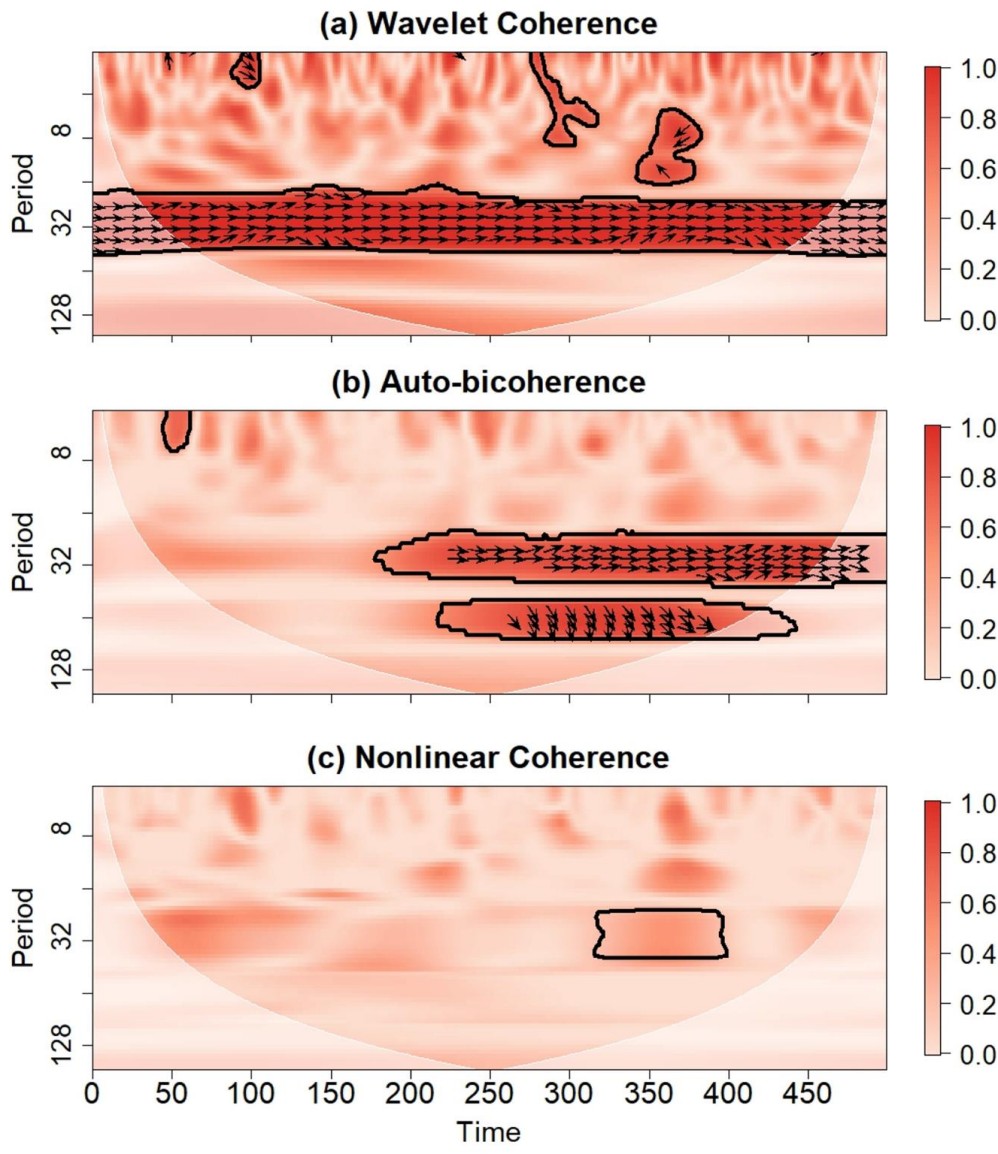


Figure 2. (a) Wavelet coherence between the time series of $F(t)$ and $R(t)$ shown in Figure 1. Arrows indicate
the relative phase difference, where arrows pointing to the right mean that the time series are in phase.
(b) The local diagonal slice of the auto-bicoherence spectrum of $F(t)$. Arrows represent the bi-phase,
where arrows pointing to the right mean that the phase coupling between the mode with period indicated
on the vertical axis and its harmonic contributes to positive skewness. (c) Nonlinear coherence between
$F(t)$ and $R(t)$. Contours in all panels enclose regions of 5% cumulative area-wise significance. Light-shaded
region represents the cone of influence where edge effects may be important.

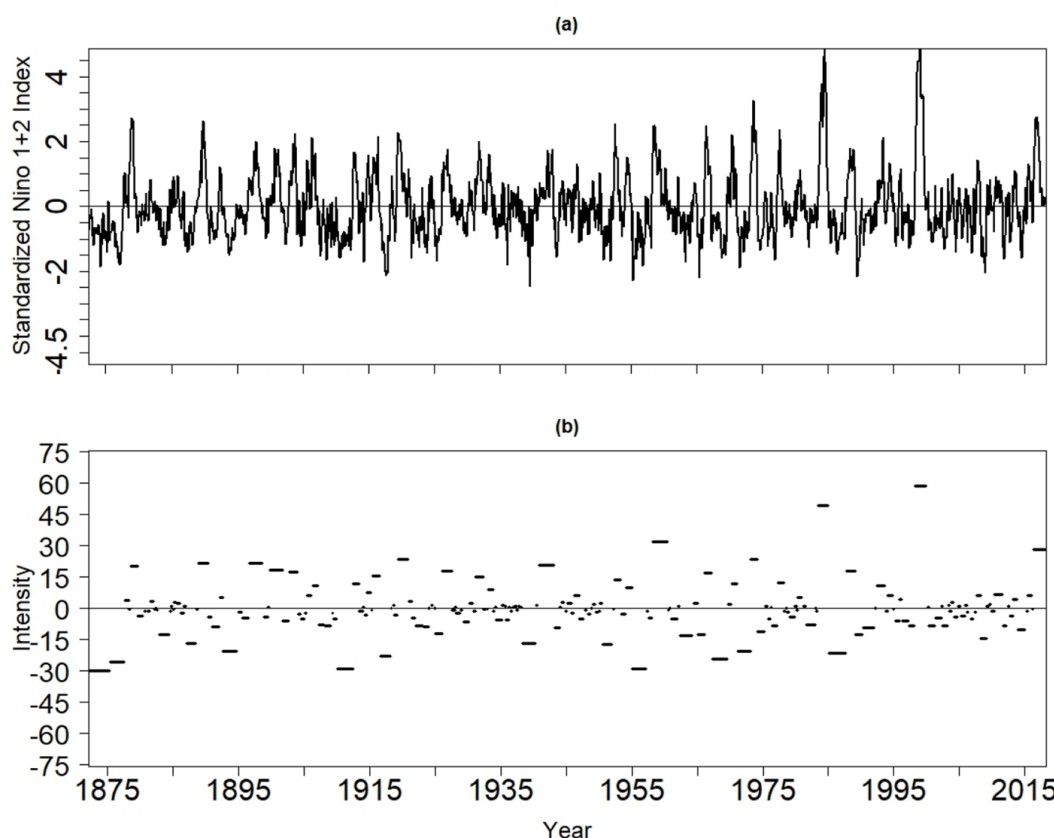

Figure 3. The (a) time series and (b) event spectrum of the Niño 1+2 index. The left and right end points of the line segments in (b) represent, respectively, the beginning and termination of events so that the length of the line segments corresponds to event persistence. The corresponding event intensity is indicated on the vertical axis.

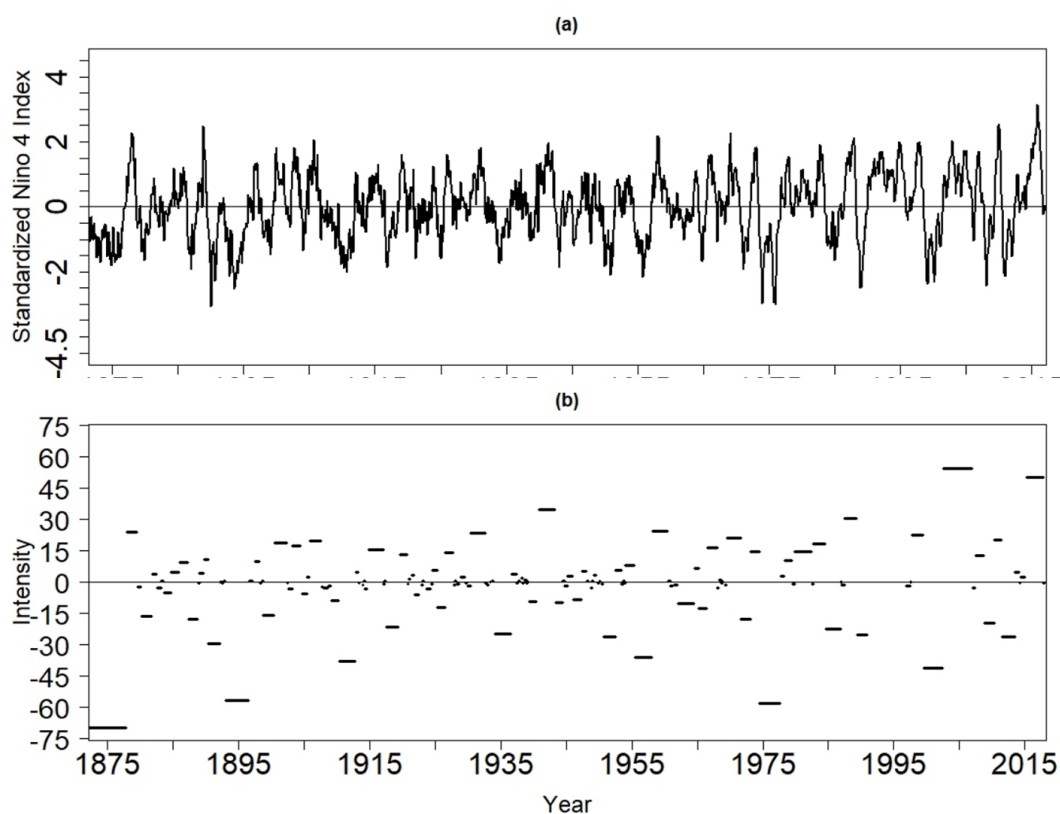


Figure 4. Same as Figure 3 but for the Niño 4 index.




Figure 5. 20-year sliding skewness of June-September All-India rainfall and time series for the Niño 1+2
and Niño 4 indices. (b) 20-year sliding correlation between anomalies for June-September All-India
rainfall and the time series for the Niño 1+2 and Niño 4 indices. (c) Same as (b) but for August-
September All-India rainfall.




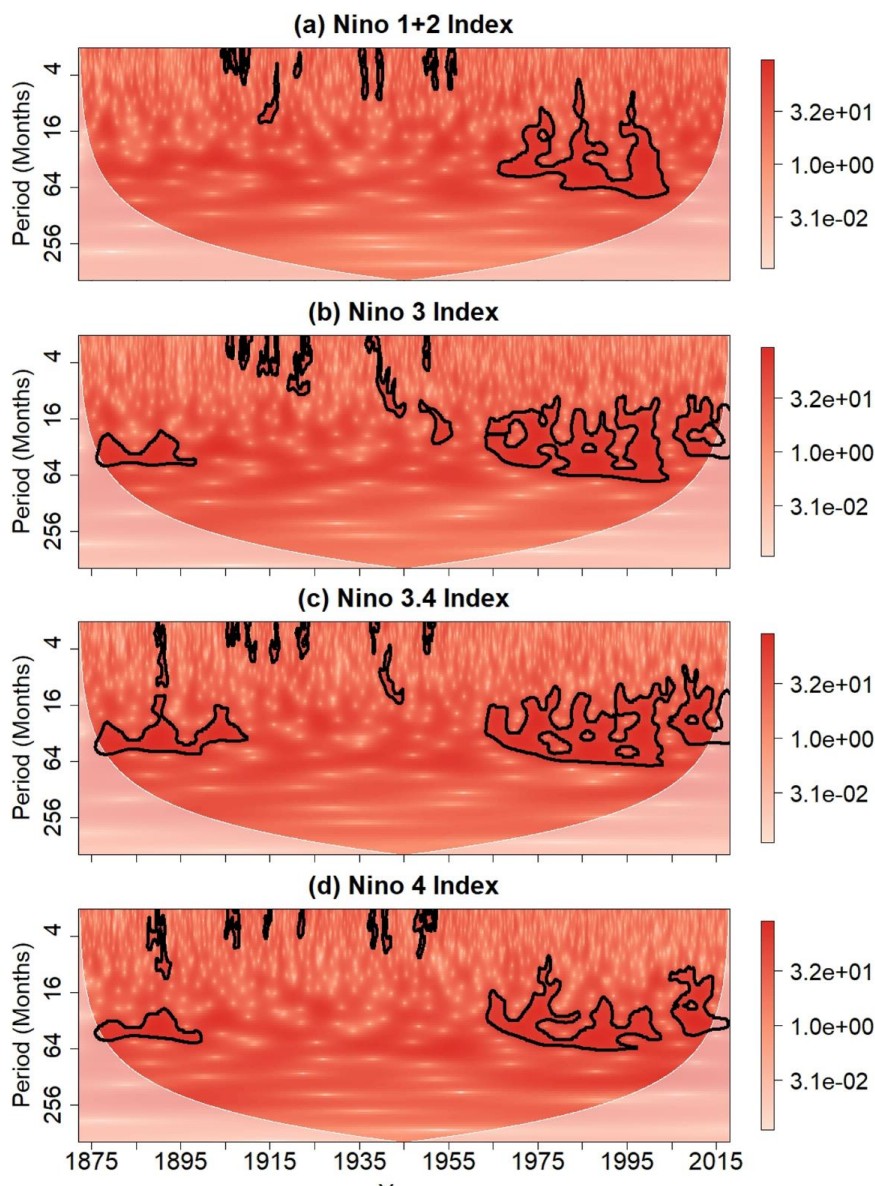


Figure 6. Wavelet Power spectrum of the (a) Niño 1+2, (b) Niño 3, (c) Niño 3.4, and (d) Niño 4 indices.
Contours enclose regions of 5% cumulative area-wise significance. Light-shaded region represents the
cone of influence, which is the region where edge effects are non-negligible.

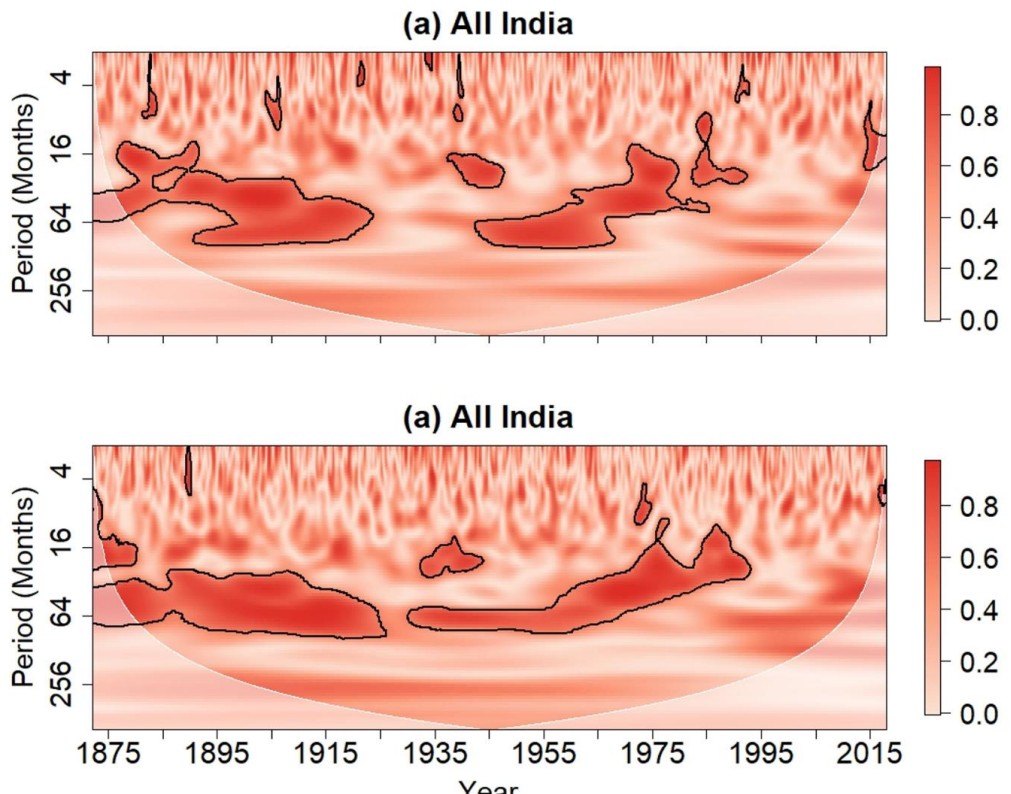

Figure 7. Wavelet coherence spectrum between All-India rainfall anomalies and time series for the (a) Niño 1+2 and (b) Niño 4 indices. Contours enclose regions of 5% cumulative area-wise significance. Light-shaded region represents the cone of influence, which is the region where edge effects are non-negligible.



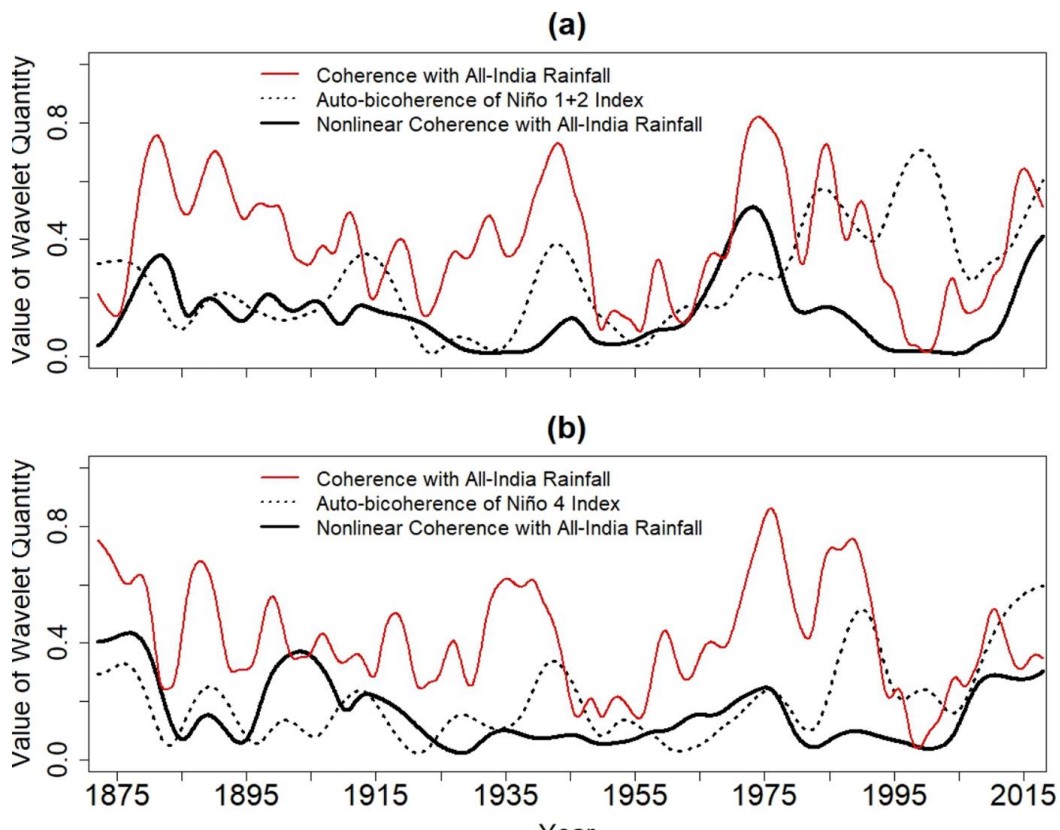

Figure 8. (a) The wavelet coherence between All-India rainfall and the Niño 1+ 2 index, the auto-
bicoherence of the Niño 1+2 index, and the nonlinear coherence between the Niño 1+2 index and All-
India rainfall anomalies averaged in the period band of 16 to 64 months. (b) The same as (a) but with the
Niño 4 index.




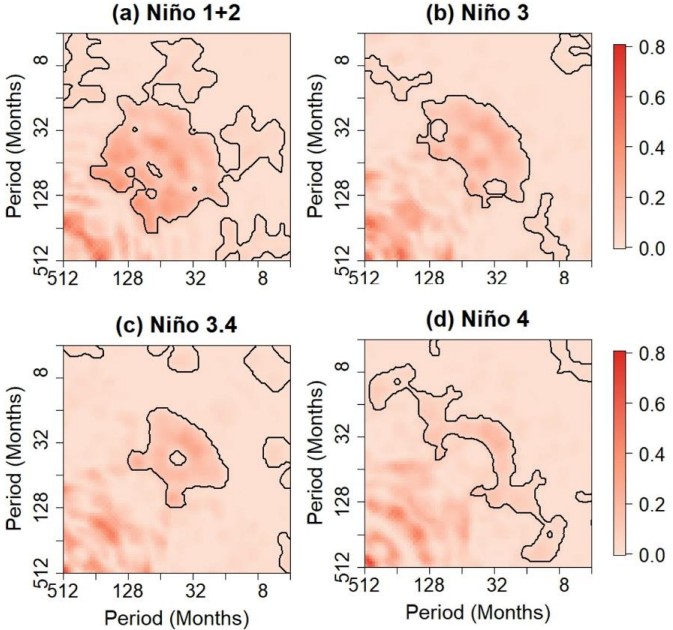


Figure 9. Global auto-bicoherence spectra of the (a) Niño 1+2, (b) Niño 3, (c) Niño 3.4, and (d) Niño 4
indices. Contours enclose regions of 5% cumulative area-wise significance.



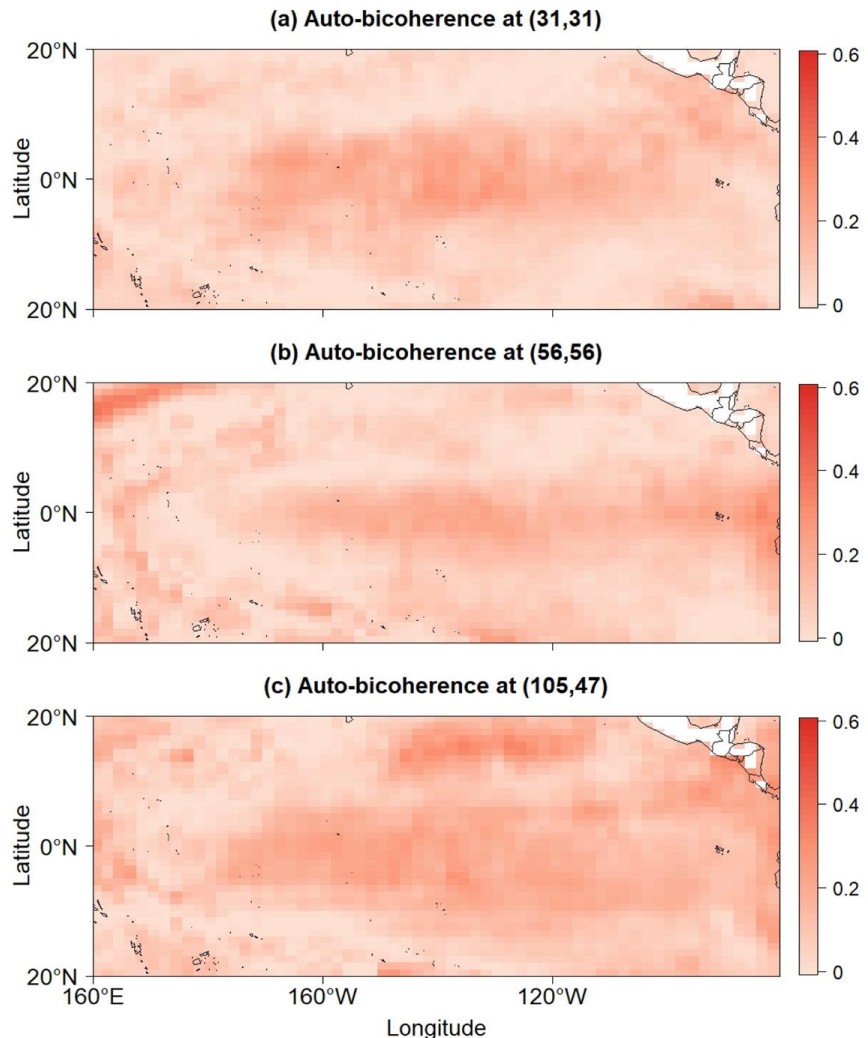


Figure 10. Global auto-bicoherence corresponding to the pairs (a) (31, 31), (b) (56, 56), and (c) (105, 47)
[months].

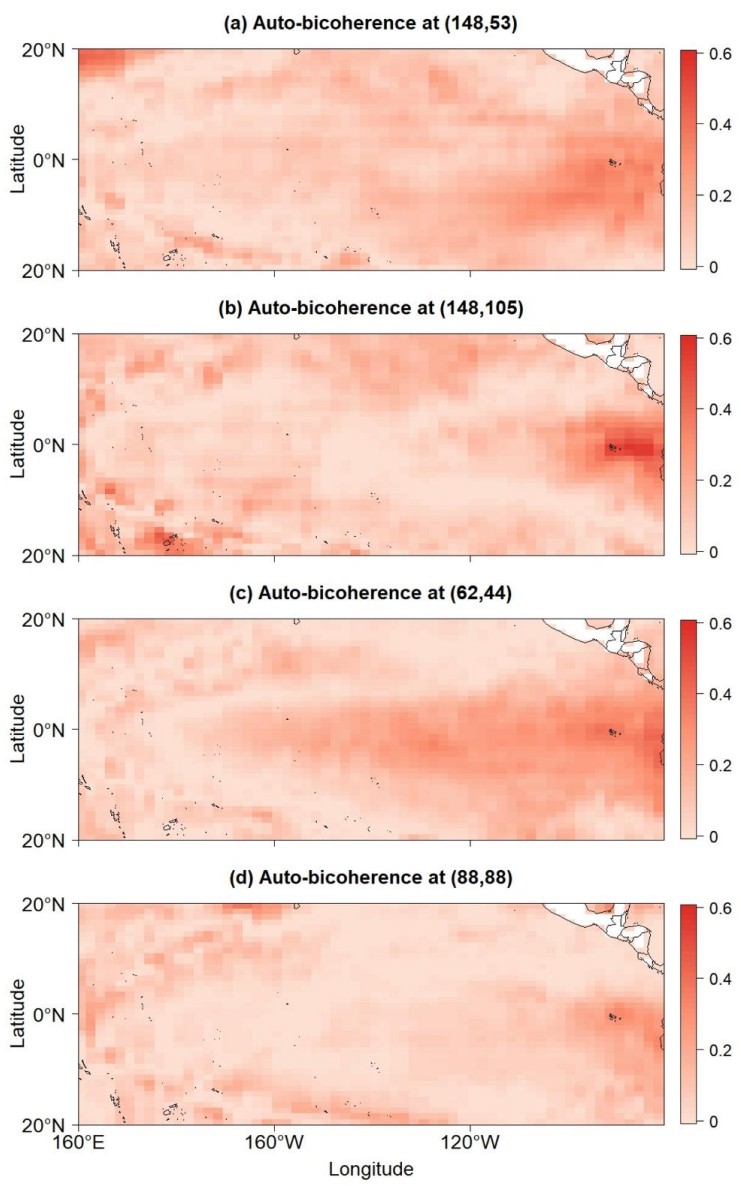


Figure 11. Global auto-bicoherence corresponding to the pairs (a) (158, 43), (b) (148, 105), (c) (62, 44),
and (d) (88,88) [months].


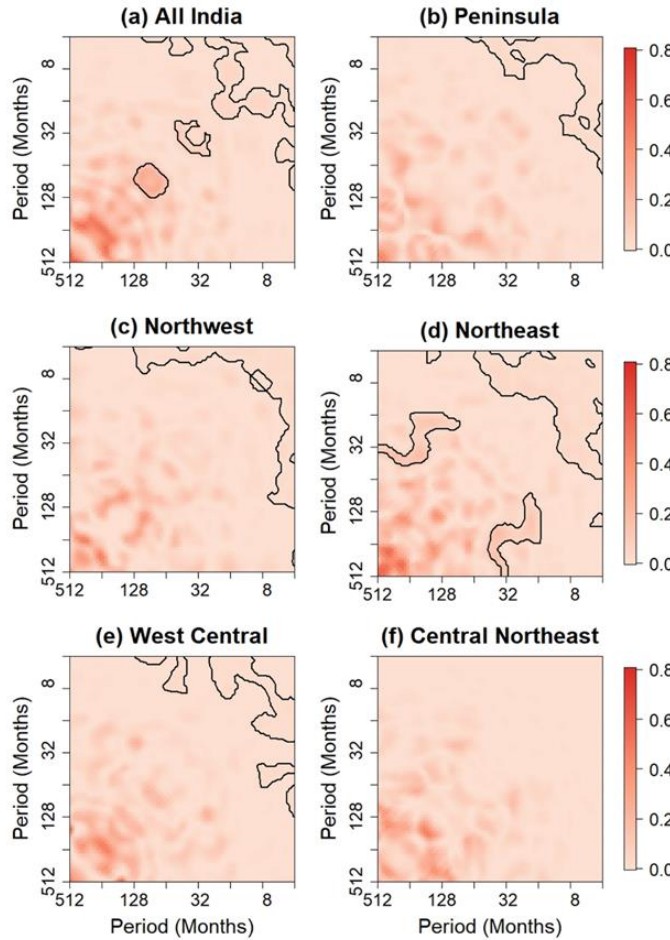

Figure 12. Global auto-bicoherence spectra of the (a) All-India, (b) Peninsula, (c) Northwest, (d) Northeast,
(e) West Central, and (f) Central Northeast time series. Contours enclose regions of 5% cumulative area-
wise significance.



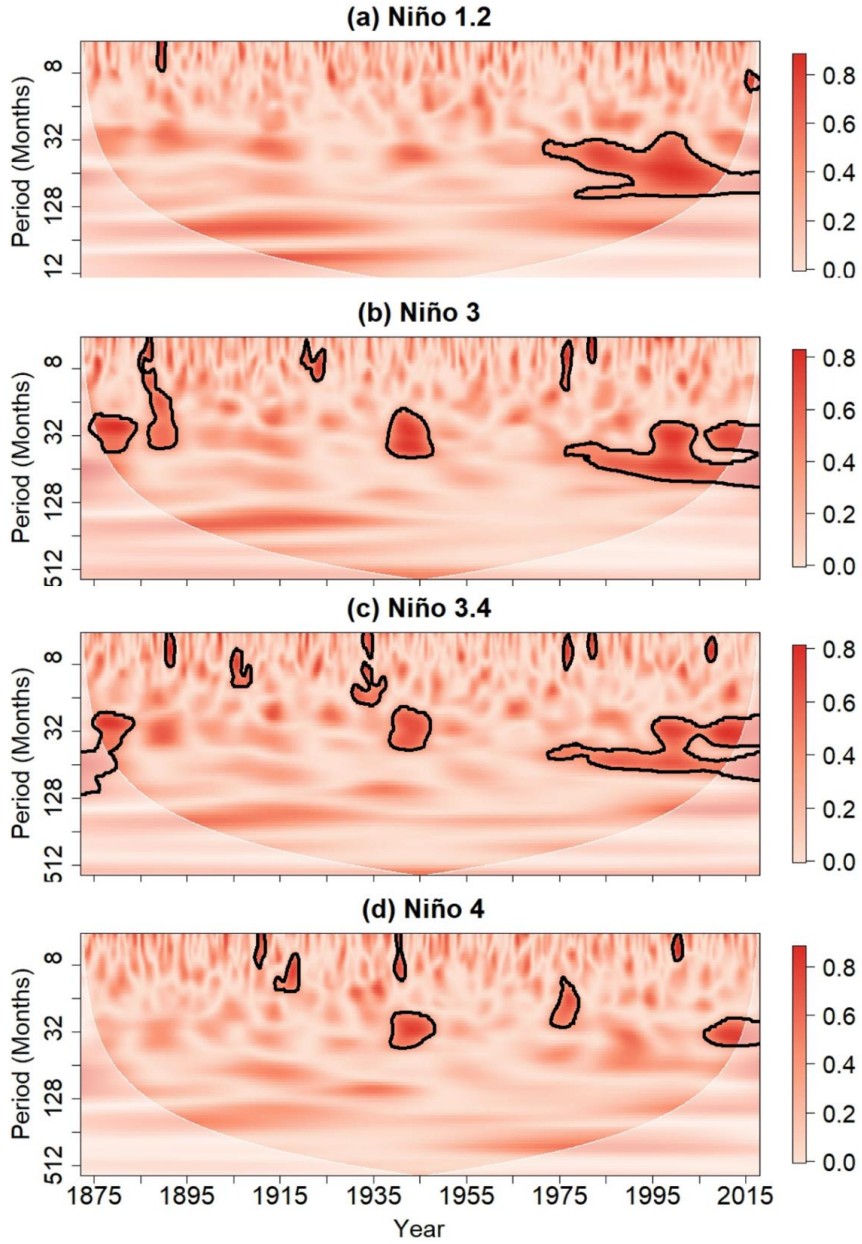


Figure 13. Local auto-bicoherence spectra of the (a) Niño 1+2, (b) Niño 3, (c) Niño 3.4, and (d) Niño 4
indices. Contours enclose regions of 5% cumulative area-wise significance and the light shading represents
the cone of influence.








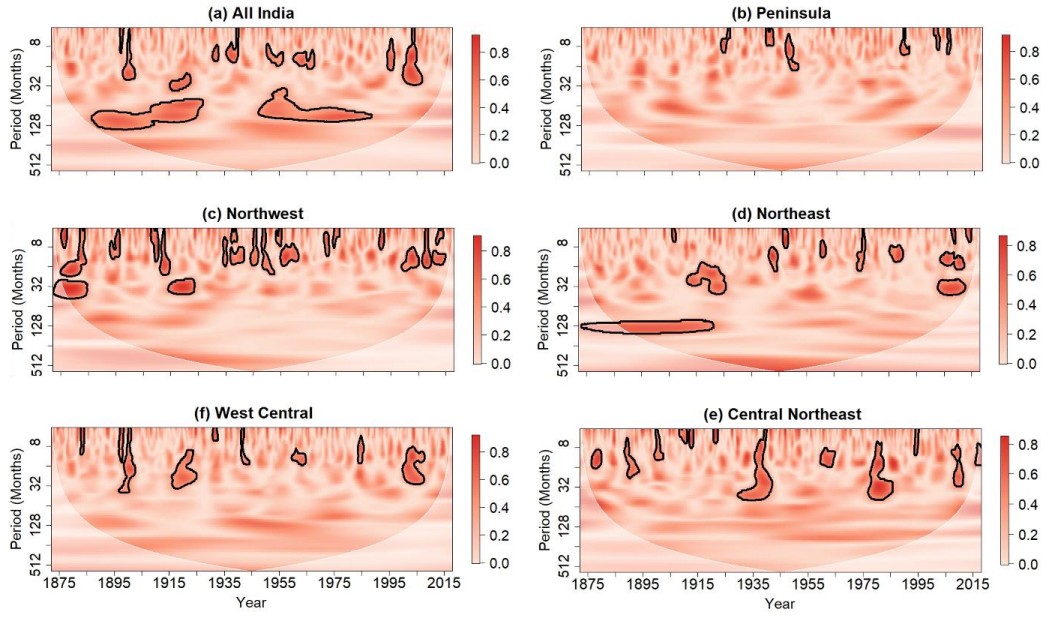


Figure 14. Local auto-bicoherence spectra of the (a) All-India, (b) Peninsula, (c) Northwest, (d) Northeast, (e) West Central, and (f) Central Northeast time series. Contours enclose regions of the 5% cumulative area-wise significance and the light shading represents the cone of influence.


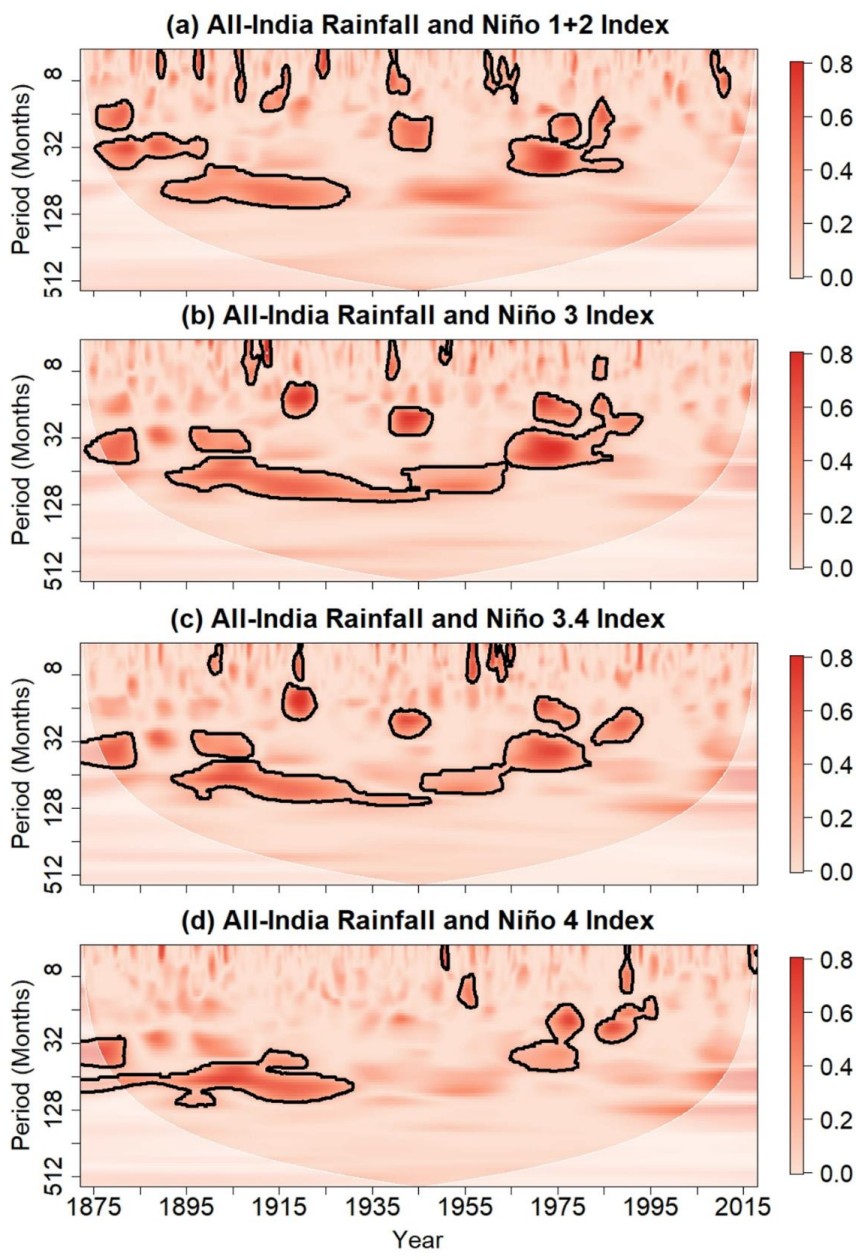


Figure 15. Nonlinear wavelet coherence between the All-India time series and times series for the (a) Niño
1+2, (b) Niño 3, (c) Niño 3.4, and (d) Niño 4 indices. Contours enclose regions of 5% cumulative area-wise
significance and light shading represents the cone of influence.
