# Peer review of "A Skewed perspective of the Indian rainfall-ENSO Relationship"

_Hydrology and Earth System Sciences, 2019_

## Referee Comment (RC1) · James Doss-Gollin (Referee) · 22 Aug 2019

The purpose of this paper is to apply novel methods for bivariate, nonlinear wavelet analysis to understand whether apparent changes in the relationship between indices for ENSO and the Indian Monsoon represent fundamental changes in their relationship. The methods are based on those published in previous peer-reviewed papers by the authors, and so this paper can be viewed as an application of these methods to a relevant and interesting scientific problem. These tools for higher-order wavelet analysis allow the authors to quantify the nonlinearity of ENSO and indices for the Indian monsoon. The authors conclude from this analysis that ENSO nonlinearity is related to ENSO flavors, and that the apparent changes in the relationship between ENSO and Indian rainfall are also related to ENSO flavors. Finally, the authors use these findings

to re-interpret findings by Yun and Timmerman (2018) which suggest that the breakdown of the ENSO-India rainfall relationship is related to shifts in the linearity of the ENSO regime. Specifically, the authors argue that the nonlinear relationship identified by their higher-order wavelet model will have non-Gaussian noise components, potentially confounding the alternative analysis. While this paper is unlikely to be the final word on this debate, it is a clear, well-written, and important contribution to the study of the ENSO-Indian rainfall relationship, and to time series analysis more broadly, and should be published pending minor stylistic edits.

I also note a lack of a data availability policy [https://www.natural-hazards-and-earth-system-sciences.net/about/data_policy.html]. Making the code and data used to others would help other researchers apply these methods to other time series.

Minor comments: please see attached PDF

Please also note the supplement to this comment:
https://www.hydrol-earth-syst-sci-discuss.net/hess-2019-280/hess-2019-280-RC1-supplement.pdf

**Supplement:**

**Comments on HESS-2019-280: A Skewed perspective of the Indian rainfall-ENSO Relationship**

August 22, 2019

The purpose of this paper is to apply novel methods for bivariate, nonlinear wavelet analysis to understand whether apparent changes in the relationship between indices for ENSO and the Indian Monsoon represent fundamental changes in their relationship. The methods are based on those published in previous peer-reviewed papers by the authors, and so this paper can be viewed as an application of these methods to a relevant and interesting scientific problem. These tools for higher-order wavelet analysis allow the authors to quantify the nonlinearity of ENSO and indices for the Indian monsoon. The authors conclude from this analysis that ENSO nonlinearity is related to ENSO flavors, and that the apparent changes in the relationship between ENSO and Indian rainfall are also related to ENSO flavors. Finally, the authors use these findings to re-interpret findings by Yun and Timmermann (2018) which suggest that the breakdown of the ENSO-India rainfall relationship is related to shifts in the linearity of the ENSO regime. Specifically, the authors argue that the nonlinear relationship identified by their higher-order wavelet model will have non-Gaussian noise components, potentially confounding the alternative analysis. While this paper is unlikely to be the final word on this debate, it is a clear, well-written, and important contribution to the study of the ENSO-Indian rainfall relationship, and to time series analysis more broadly, and should be published pending minor stylistic edits.

I also note a lack of a data availability policy.[1] Making the code and data used to others would help other researchers apply these methods to other time series.

**Specific comments**

- L9: It took me a while to understand the similarities and differences between the terms auto-bicoherence, bicoherence, coherence, etc. The auto-bicoherence is defined later, but perhaps a simple table or sentence near the introduction explaining the difference between these different terms would be helpful. (I am flagging this in the abstract but the clarification could happen elsewhere)

- L48: consider rephrasing "investigators"

- LL58-59: there are also concerns about data quality – would be worth at least referencing or discussing them
* * *
[1] https://www.natural-hazards-and-earth-system-sciences.net/about/data_policy.html

- L95 and beyond: please consider converting from month$^{-1}$ to year$^{-1}$

- L117: are there possible data quality issues with the rainfall data?

- L135: the formatting here has changed

- L146: are there cases where very small events (say a single month) emerge? If so how are these handled?

- L156: Consider re-wording to continuous wavelet transform of a time series $X = \dots$ as a function of wavelet scale $s$ is given by

- L160: if this transform is commonly used please cite. Are results sensitive to choice of wavelet form or to choice of $\omega$?

- L188: it would help to be clearer here about what sorts of nonlinearities this analysis can pick up, which sorts of nonlinearities it cannot pick up, and what sorts of nonlinearities have been hypothesized or observed in ENSO time series.

- L200: see above comment regarding distinction between coherence, auto-coherence, etc.

- L464: if there are spatial shifts happening that are related to ENSO, this could potentially complicate some of this analysis correct?

- L545: Consider re-wording "despite how"

- L595: what is your interpretation of the finding that the modes found are not harmonics of 12 months? Given that the seasonal (12 month) cycle is important here and many of the other modes may be coupled to it, it would be useful to explain to the reader why other modes emerge as important.

- L610: this is an important point which the authors should consider emphasizing in the abstract

- Figure 5: consider adding color

- Figure 6: consider plotting the global (average) wavelet spectrum adjacent

- Figure 7: please fix titles

- Figure 8: the figure has gotten clipped at the left margin

- Figure 9: this is the wrong place to bring this up but it would be helpful to add some discussion in the methods section, specifically around hypothesis testing, about what the 5% cumulative area-wise significance means and how to interpret it.

- Figure 11: please clarify why these pairs were chosen

**References**

Yun, Kyung-Sook and Axel Timmermann (2018). "Decadal Monsoon-ENSO Relationships Reexamined". *Geophysical Research Letters* 45.4. DOI: 10.1002/2017GL076912.

---

## Referee Comment (RC2) · Anonymous Referee #2 · 10 Sep 2019

GENERAL COMMENTS The manuscript investigates the relationship between rainfall and ENSO index using wavelet analysis and a wavelet coherence method is proposed to explain the changes of temporal correlation between two time series. The topic of the study is interesting may be outside the scope of the journal. This appears more so as almost all references are from climate journals where this paper sits more naturally. I think this paper should be withdrawn and submitted to an appropriate climate journal, or else reformatted to represent better arguments as to why it is of interest to hydrology directly. I have, however, read through the paper and have some comments that may help the authors publish this succesfully.

COMMENT1: Section 2, the way authors computed the monthly anomaly by subtracting the data from the whole period is not the recommended and standard way. It is

recommended by WMO that a fixed reference period is defined as the 30-year period 1 January 1961 to 31 December 1990. Authors should consider use this as baseline period, especially when compared SST and Nino indices over different regions. COMMENT2: Section 3.1, there is no such Reference Schulte and Lee (2019). More importantly, the reason of adapting Event Decomposition is not well explained and how it helps quantifying the nonlinearity (i.e. skewness) of rainfall and ENSO index is not demonstrated. In the end of Method section, authors considered a synthetic example to illustrate the concept of nonlinear coherence using original time series but following your methodology it should be transformed to event spectra before calculating the coherence. The impact of Event Decomposition on the wavelet analysis and coherency is unknown. COMMENT3: Section 4.2, the relation between skewness and correlation is not explicitly demonstrated. There is a sharp decrease of skewness of June-September rainfall around 1991. Is there any particular reason? And what is the implication of this change? "A comparison of Figures 5a and Figures 5c reveals that the weakening and reversal of the relationship occurs during the time period when the Niño 1+2 index is especially skewed, suggesting that ENSO skewness changes could be contributing to changes in the time-domain correlation between ENSO and All-India rainfall. " This conclusion is in doubt, Figure 5a doesn't include the skewness of August-September rainfall. COMMENT4: Section 4.2 and 4.3, through the global and local auto-bicoherence analysis, they show the nonlinearity of ENSO indices and India rainfall in the frequency and spatial space individually. But how these two related to each other, authors do not explain explicitly. Using the nonlinear wavelet coherence method to test your hypothesis should be the major contribution of your work, however it is only briefly discussed in the very end of Section 4.3.2. There are lots of redundant information in the manuscript, which makes the paper long and difficult to read.

SPECIFIC POINTS: 1. EL Nino or El Nino, please keep it consistent throughout the paper. 2. Line 104, keep the numbering format consistent. 3. Please have a careful look of the format of your references. 4. Line 274, keep the equation numbering format consistent. 5: Line 175, Because theory supports a casual link. . .Authors do

not explain this point well. Does strong coherence or association mean causality in nonlinear system? More details are needed. 6. Figure 7, what monsoon rainfall is used, full monsoon or late monsoon? 7. Line 447, what is the abbreviation of AIR standing for? 8. Line 135-137, keep the font format consistent.

I recommend authors to do a search on [hydrology and wavelets and precipitation and "el Nino"] or maybe "low frequency variability" and see how they have established the link of their paper to the hydrology audience they are presenting to. It may give authors a good idea of how they could improve their pitch.

---

## Author Comment (AC1) · 28 Oct 2019

**Summary**

The authors appreciate the many detailed suggestions. They will be incorporated into the revised manuscript. The revised manuscript will include refined figures, less redundant information, and references to hydrological studies that will highlight the importance of the study to the hydrology and earth system science community. The revised manuscript will also include a URL to the first author's website where the computer software for the adopted methodology can be obtained. More detailed responses to comments are provided below. The reviewer comments are in bold text and our responses to the comments are in plain text.

**Reviewer 1**

**The purpose of this paper is to apply novel methods for bivariate, nonlinear wavelet analysis to understand whether apparent changes in the relationship between indices for ENSO and the Indian Monsoon represent fundamental changes in their relationship. The methods are based on those published in previous peer-reviewed papers by the authors, and so this paper can be viewed as an application of these methods to a relevant and interesting scientific problem. These tools for higher-order wavelet analysis allow the authors to quantify the nonlinearity of ENSO and indices for the Indian monsoon. The authors conclude from this analysis that ENSO nonlinearity is related to ENSO flavors, and that the apparent changes in the relationship between ENSO and Indian rainfall are also related to ENSO flavors. Finally, the authors use these findings to re-interpret findings by Yun and Timmerman (2018) which suggest that the breakdown of the ENSO-India rainfall relationship is related to shifts in the linearity of the ENSO regime. Specifically, the authors argue that the nonlinear relationship identified by their higher-order wavelet model will have non-Gaussian noise components, potentially confounding the alternative analysis. While this paper is unlikely to be the final word on this debate, it is a clear, well-written, and important contribution to the study of the ENSO-Indian rainfall relationship, and to time series analysis more broadly, and should be published pending minor stylistic edits. I also note a lack of a data availability policy [https://www.natural-hazards-and-earthsystem-sciences.net/about/data_policy.html]. Making the code and data used to others would help other researchers apply these methods to other time series.**

The revised manuscript will include a URL to the first author's website where the computer software for the adopted methodology can be obtained. A link to the data sets used will also be provided.

**L9: It took me a while to understand the similarities and differences between the terms auto-bicoherence, bicoherence, coherence, etc. The auto-bicoherence is defined later, but perhaps a simple table or sentence near the introduction explaining the difference between these different terms would be helpful. (I am flagging this in the abstract but the clarification could happen elsewhere)**

To better distinguish coherence traditionally used from the new methods, traditional coherence will be referred to as "linear coherence" in the revised abstract. Furthermore, a sentence mentioning how auto-bicoherence detects quadratic nonlinearities in time series will also be added to the abstract and main text. The type of nonlinearities (e.g. cubic nonlinearities) that cannot be detected by the methods will also be discussed in the methods section. A table will be added to the revised manuscript to clarify the nomenclature used in the paper.

**L48: consider rephrasing "investigators"**

"investigators" will be changed to "researchers" in the revised manuscript.

**L58-59: there are also concerns about data quality – would be worth at least referencing or discussing them https://www.natural-hazards-and-earth-system-sciences.net/about/data_policy.html**

A careful literature search did not reveal any studies citing data quality issues regarding the All-India rainfall (AIR) data set. Nevertheless, as a gauge-based product, AIR has both the advantages and disadvantages of any product based on in situ weather stations—that is, the data that go into it are collected in a well-understood way, without use of proxies, but there is the potential for non-representative station distribution or faulty gauges. That said, AIR is widely used product that has been applied successfully to many studies of weather and climate in India.

**L95 and beyond: please consider converting from month−1 to year−1**

After careful consideration, we decided that we will still use months in the revised manuscript because of months seems to work better with how the wavelet scales and periods are calculated using powers of 2.

**L117: are there possible data quality issues with the rainfall data?**

To the authors knowledge, there are no serious data quality issues. The All-India rainfall data is frequently used in Indian Monsoon studies. Because of the importance of the Indian monsoon, careful data collection has been conducted since the 1800's.

**L135: the formatting here has changed.**

Thank you for identifying the formatting change. It will be corrected in the revised manuscript.

**L146: are there cases where very small events (say a single month) emerge? If so how are these handled?**

To make the manuscript more concise, the event decomposition approach will be removed from the manuscript. Nevertheless, the single-month events were consider to be short events whose intensities were the values of the data points composing the events.

**L156: Consider re-wording to continuous wavelet transform of a time series X = ... as a function of wavelet scale s is given by**

The authors appreciate the suggestion for rewording the sentence. The sentence will be reworded in the revised manuscript.

**L160: if this transform is commonly used please cite. Are results sensitive to choice of wavelet form or to choice of ω?**

The Morlet wavelet is the most commonly used analyzing wavelet in climate and hydrological studies because it balances time and frequency localization. We will add a citation to reflect the common use of the Morlet wavelet. The choice of $\omega$ would impact the results given that it alters the time and frequency localization behavior of the Morlet wavelet. However, given that $\omega = 6$ is such a common choice in wavelet applications, the authors feel that it is beyond the scope of the paper to understand its effect on the interpretability of wavelet analysis results.

**L188: it would help to be clearer here about what sorts of nonlinearities this analysis can pick up, which sorts of nonlinearities it cannot pick up, and what sorts of nonlinearities have been hypothesized or observed in ENSO time series.**

We agree that a clarification is needed. In the revised manuscript, it will be mentioned that the bicoherence method detects quadratic type nonlinearities.

**L200: see above comment regarding distinction between coherence, auto-coherence, etc.**

A table will be added to the revised manuscript to clarify the nomenclature used in the paper.

**L464: if there are spatial shifts happening that are related to ENSO, this could potentially complicate some of this analysis correct?**

While we agree that spatial patterns could be shifting, the purpose of the analysis was to quantify the auto-bicoherence of SSTs at various grid points. An additional study would be needed to see if there are spatial shifts in the patterns, which is beyond the scope of the paper.

**L545: Consider re-wording "despite how"**

"Despite how" will be reworded in the revised manuscript**.**

**L595: what is your interpretation of the finding that the modes found are not harmonics of 12 months? Given that the seasonal (12 month) cycle is important here and many of the other modes may be coupled to it, it would be useful to explain to the reader why other modes emerge as important.**

Although understanding the specific dynamics underlying the nonlinear modes is beyond the scope of paper, the revised manuscript will include references to studies that focused on understanding nonlinear ENSO dynamics.

**L610: this is an important point which the authors should consider emphasizing in the abstract**

The authors agree that it is important point. As such, the finding will be discussed in the revised abstract.

**Figure 5: consider adding color**

We agree that this figure could be clearer. To remedy the drawback, line weights and styles will be changed in the revised manuscript.

**Figure 6: consider plotting the global (average) wavelet spectrum adjacent**

The authors agree that the global wavelet power spectra would help highlight the dominant peaks in the wavelet power spectra. Therefore, global wavelet power spectra will be placed adjacent to the full wavelet power spectrum in the revised manuscript.

**Figure 7: please fix titles**

Thank you for referring us to the errors in the titles. The problem will be corrected in the revised manuscript.

**Figure 8: the figure has gotten clipped at the left margin.**

Thank you for referring us to the clipping problem. The problem will be corrected in the revised manuscript.

**Figure 9: this is the wrong place to bring this up but it would be helpful to add some discussion in the methods section, specifically around hypothesis testing, about what the 5% cumulative area-wise significance means and how to interpret it.**

The authors agree that some discussion is warranted given the novelty of the statistical tests. A short discussion will be added in the revised manuscript, but the reader will be referred to Schulte (2019) for more details.

**Figure 11: please clarify why these pairs were chosen**

As stated on Line 481, the pairs were chosen because they are local maxima in auto-bicoherence that are statistically significant. Choosing local maxima allows the spatial patterns shown in Figure 11 to emerge more clearly. A sentence clarifying our choice of pairs will be inserted into the revised manuscript.

**References**

**References Yun, Kyung-Sook and Axel Timmermann (2018). "Decadal Monsoon-ENSO Relationships Reexamined". Geophysical Research Letters 45.4. doi: 10.1002/2017GL076912.**

---

## Author Comment (AC2) · 28 Oct 2019

**Summary**

The authors appreciate the many detailed suggestions. They will be incorporated into the revised manuscript. The revised manuscript will include refined figures, less redundant information, and references to hydrological studies that will highlight the importance of the study to the hydrology and earth system science community. The revised manuscript will also include a URL to the first author's website where the computer software for the adopted methodology can be obtained. More detailed responses to comments are provided below. The reviewer comments are in bold text and our responses to the comments are in plain text.

**GENERAL COMMENTS The manuscript investigates the relationship between rainfall and ENSO index using wavelet analysis and a wavelet coherence method is proposed to explain the changes of temporal correlation between two time series. The topic of the study is interesting may be outside the scope of the journal. This appears more so as almost all references are from climate journals where this paper sits more naturally. I think this paper should be withdrawn and submitted to an appropriate climate journal, or else reformatted to represent better arguments as to why it is of interest to hydrology directly. I have, however, read through the paper and have some comments that may help the authors publish this successfully.**

Our manuscript addresses precipitation variability in the South Asian monsoon region. As precipitation in this region is critical to mountain glaciers, transboundary rivers, groundwater recharge, and functioning of many ecosystems and human systems, we believe that the topic is of considerable importance to hydrologists and Earth system scientists. However, we appreciate the reviewer's concern, and we recognize that we did not sufficiently emphasize the hydrological relevance of this work in the original manuscript. In the revised manuscript we will elaborate on both the hydrological motivation for our study and the hydrological implications of our results.

**COMMENT1: Section 2, the way authors computed the monthly anomaly by subtracting the data from the whole period is not the recommended and standard way. It is recommended by WMO that a fixed reference period is defined as the 30-year period 1 January 1961 to 31 December 1990. Authors should consider use this as baseline period, especially when compared SST and Nino indices over different regions.**

Although the authors agree that a 30-year base period is commonly used in climate studies, it is unclear how using the standard base period would benefit the present analysis. Using a different base period would only translate the time series up or down uniformly and would not alter the actual behavior of the time series. Therefore, the results of the wavelet analysis, which are the focus of the present study, would be unchanged. Furthermore, skewness would be made relative to that base period, which would make the more recent skewed events appear less prominent. As such, we feel that subtracting the long-term means to calculate anomalies is the best approach for the present study. By using long-term means, it is easier to see how skewness how evolved throughout the study period, which is consistent with what the wavelet analysis is quantifying.

**COMMENT2: Section 3.1, there is no such Reference Schulte and Lee (2019). More importantly, the reason of adapting Event Decomposition is not well explained and how it helps quantifying the nonlinearity (i.e. skewness) of rainfall and ENSO index is not demonstrated. In the end of Method section, authors considered a synthetic example to illustrate the concept of nonlinear coherence using**

**original time series but following your methodology it should be transformed to event spectra before calculating the coherence. The impact of Event Decomposition on the wavelet analysis and coherency is unknown.**

The authors agree that it is unclear how the event spectra benefit the paper. As such, we will remove the event decomposition method results from the revised manuscript. This removal will allow us to focus more on wavelet analysis, which is the main topic of the paper. The authors note that the coherence spectra are based on the actual time series and not on the event transformed time series.

**COMMENT3: Section 4.2, the relation between skewness and correlation is not explicitly demonstrated. There is a sharp decrease of skewness of June-September rainfall around 1991. Is there any particular reason? And what is the implication of this change? "A comparison of Figures 5a and Figures 5c reveals that the weakening and reversal of the relationship occurs during the time period when the Niño 1+2 index is especially skewed, suggesting that ENSO skewness changes could be contributing to changes in the time-domain correlation between ENSO and All-India rainfall. " This conclusion is in doubt, Figure 5a doesn't include the skewness of August-September rainfall.**

The main reason for showing skewness is because two time series can only be perfectly correlated if all the statistical moments are correlated. More specifically, if the skewness of one time series is increasing but another remains nearly constant, then the lack of correlation between skewness of the two time series must be contributing to changes in the correlation between the time series. This idea will be made more explicit in the revised manuscript. The sharp decrease in skewness around 1991 could be noise. A full understanding of all sources of Indian rainfall skewness would require additional analyses, which would digress from the focus of the paper, which is to relate ENSO skewness to Indian rainfall skewness. Nevertheless, an implication of the result is that noise also influences the correlation between Indian rainfall and ENSO. This possibility will be mentioned in the revised manuscript.

**COMMENT4: Section 4.2 and 4.3, through the global and local auto-bicoherence analysis, they show the nonlinearity of ENSO indices and India rainfall in the frequency and spatial space individually. But how these two related to each other, authors do not explain explicitly.**

The main reasons for showing the local and global bicoherence analyses is to highlight their differences. Because the nonlinear coherence between them is weak, we expected that the differences to be large. This point will be clarified in the revised manuscript.

**Using the nonlinear wavelet coherence method to test your hypothesis should be the major contribution of your work, however it is only briefly discussed in the very end of Section 4.3.2. There are lots of redundant information in the manuscript, which makes the paper long and difficult to read.**

In the revised manuscript, we will expand the nonlinear wavelet coherence method section. In addition, some text will be moved or deleted so that the nonlinear coherence section appears earlier in the revised manuscript. The authors agree that there is a lot of redundant information. The authors will remove the redundant information in the revised manuscript by deleting text and moving some information to the supplementary material.

**SPECIFIC POINTS:**

**1. EL Nino or El Nino, please keep it consistent throughout the paper.**

The authors appreciate the identification of this inconsistency, which will be corrected throughout the paper. The format will be changed to "El Nino" throughout.

**2. Line 104, keep the numbering format consistent.**

The numbering inconsistency will be corrected in the revised manuscript.

**3. Please have a careful look of the format of your references.**

The authors appreciate the comment about the reference formatting. The reference formatting will be corrected in the revised manuscript.

**4. Line 274, keep the equation numbering format consistent.**

The equation formatting inconsistency will be corrected in the revised manuscript.

**5: Line 175, Because theory supports a casual link...Authors do not explain this point well. Does strong coherence or association mean causality in nonlinear system? More details are needed.**

Authors agree that more details are needed regarding the causal linkage statement. There are many studies that have linked ENSO to the Indian monsoon These studies will be referenced in the revised manuscript. In addition, a few sentences will be added to describe how ENSO is physically related to the Indian Monsoon.

**6. Figure 7, what monsoon rainfall is used, full monsoon or late monsoon?**

The authors note that Figure 7 is the wavelet power spectrum of the All-India time series without any seasonal averaging.

**7. Line 447, what is the abbreviation of AIR standing for?**

The abbreviation stands for All-India Rainfall. The abbreviation will be introduced when the All-India rainfall data are first mentioned.

**8. Line 135-137, keep the font format consistent.**

The authors appreciate the identification of the formatting issue, which will be corrected in the revised manuscript.

**I recommend authors to do a search on [hydrology and wavelets and precipitation and "el Nino"] or maybe "low frequency variability" and see how they have established the link of their paper to the hydrology audience they are presenting to. It may give authors a good idea of how they could improve their pitch.**

The authors appreciate the suggestion regarding a literature search.  We will conduct a literature search on wavelet analysis and hydrology and include many new references into the revised manuscript. More specifically, we well include references to papers in which wavelet coherence was used to understand hydrological processes.

---

## Author Response (AR1)

**Summary**

The authors appreciate the many detailed suggestions, which were incorporated into the revised manuscript. The
revised manuscript includes refined figures, less redundant information, and references to hydrological studies that
highlight the importance of the study to the hydrology and earth system science community. Some sections of the
original manuscript have been entirely deleted. The deleted sections are those examining the auto-bicoherence of
Indian rainfall and the event decomposition of time series. The abstract and the first two paragraphs of the introduction
section have been rewritten to better reflect the readership of the journal. The revised manuscript also includes an
URL at the end of the discussion section to direct the reader to the computer software associated with the adopted
methodology. More detailed responses to comments are provided below. The reviewer comments are in bold text and
our responses to the comments are in plain text.

**Reviewer 1**

**The purpose of this paper is to apply novel methods for bivariate, nonlinear wavelet analysis to**
**understand whether apparent changes in the relationship between indices for ENSO and the Indian**
**Monsoon represent fundamental changes in their relationship. The methods are based on those**
**published in previous peer-reviewed papers by the authors, and so this paper can be viewed as an**
**application of these methods to a relevant and interesting scientific problem. These tools for higher-**
**order wavelet analysis allow the authors to quantify the nonlinearity of ENSO and indices for the Indian**
**monsoon. The authors conclude from this analysis that ENSO nonlinearity is related to ENSO flavors,**
**and that the apparent changes in the relationship between ENSO and Indian rainfall are also related to**
**ENSO flavors. Finally, the authors use these findings to re-interpret findings by Yun and Timmerman**
**(2018) which suggest that the breakdown of the ENSO-India rainfall relationship is related to shifts in**
**the linearity of the ENSO regime. Specifically, the authors argue that the nonlinear relationship**
**identified by their higher-order wavelet model will have non-Gaussian noise components, potentially**
**confounding the alternative analysis. While this paper is unlikely to be the final word on this debate, it**
**is a clear, well-written, and important contribution to the study of the ENSO-Indian rainfall relationship,**
**and to time series analysis more broadly, and should be published pending minor stylistic edits. I also**
**note a lack of a data availability policy [https://www.natural-hazards-and-earthsystem-**
**sciences.net/about/data_policy.html]. Making the code and data used to others would help other**
**researchers apply these methods to other time series.**

A URL to the first author's website where the computer software for the adopted methodology can be
obtained is provided Page 12 of the revised manuscript.

**L9: It took me a while to understand the similarities and differences between the terms auto-**
**bicoherence, bicoherence, coherence, etc. The auto-bicoherence is defined later, but perhaps a simple**
**table or sentence near the introduction explaining the difference between these different terms would**
**be helpful. (I am flagging this in the abstract but the clarification could happen elsewhere)**

To better distinguish coherence traditionally used from the new methods, traditional coherence is now
referred to as "linear coherence" in the revised abstract. Furthermore, a sentence mentioning how auto-
bicoherence detects quadratic nonlinearities in time series has been added to the main text on Page 4
Line 174. The type of nonlinearities (e.g. cubic nonlinearities) that cannot be detected by the methods is
mentioned on Page 5 Line 185. A table (Table 1 in revised manuscript) was added to clarify the
nomenclature used in the paper.

**L48: consider rephrasing "investigators"**

"investigators" was changed to "researchers" in the revised manuscript.

**L58-59: there are also concerns about data quality – would be worth at least referencing or discussing**
**them https://www.natural-hazards-and-earth-system-sciences.net/about/data_policy.html**

A careful literature search did not reveal any studies citing data quality issues regarding the All-India
rainfall (AIR) data set. Nevertheless, as a gauge-based product, AIR has both the advantages and
disadvantages of any product based on in situ weather stations—that is, the data that go into it are
collected in a well-understood way, without use of proxies, but there is the potential for non-
representative station distribution or faulty gauges. That said, AIR is widely used product that has been
applied successfully to many studies of weather and climate in India.

**L95 and beyond: please consider converting from month−1 to year−1**

After careful consideration, we decided that we will still use months in the revised manuscript because of
months seems to work better with how the wavelet scales and periods are calculated using powers of 2.

**L117: are there possible data quality issues with the rainfall data?**

To the authors knowledge, there are no serious data quality issues. The All-India rainfall data is frequently
used in Indian Monsoon studies. Because of the importance of the Indian monsoon, careful data collection
has been conducted since the 1800's.

**L135: the formatting here has changed.**

Thank you for identifying the formatting change. It was corrected in the revised manuscript.

**L146: are there cases where very small events (say a single month) emerge? If so how are these**
**handled?**

To make the manuscript more concise, the event decomposition approach will be removed from the
manuscript. Nevertheless, the single-month events were considered to be short events whose intensities
were the values of the data points composing the events.

**L156: Consider re-wording to continuous wavelet transform of a time series X = ... as a function of**
**wavelet scale s is given by**

The authors appreciate the suggestion for rewording the sentence. The sentence was recorded in the
revised manuscript (Page 3, Line 136 of revised manuscript).

**L160: if this transform is commonly used please cite. Are results sensitive to choice of wavelet form or**
**to choice of ω?**

The Morlet wavelet is the most commonly used analyzing wavelet in climate and hydrological studies
because it balances time and frequency localization. A citation was added to indicate the common use of
the Morlet wavelet on Page 4 Line 142. The choice of $\omega$ would impact the results given that it alters the
time and frequency localization behavior of the Morlet wavelet. However, given that $\omega = 6$ is such a
common choice in wavelet applications, the authors feel that it is beyond the scope of the paper to
understand its effect on the interpretability of wavelet analysis results.

**L188: it would help to be clearer here about what sorts of nonlinearities this analysis can pick up, which sorts of nonlinearities it cannot pick up, and what sorts of nonlinearities have been hypothesized or observed in ENSO time series.**

We agree that a clarification is needed. In the revised manuscript, we now mention that the bicoherence method detects quadratic type nonlinearities on on Page 4 Line 174.

**L200: see above comment regarding distinction between coherence, auto-coherence, etc.**

A table (Table 1) was added to the revised manuscript to clarify the nomenclature used in the paper.

**L464: if there are spatial shifts happening that are related to ENSO, this could potentially complicate some of this analysis correct?**

While we agree that spatial patterns could be shifting, the purpose of the analysis was to quantify the auto-bicoherence of SSTs at various grid points. An additional study would be needed to see if there are spatial shifts in the patterns, which is beyond the scope of the paper.

**L545: Consider re-wording "despite how"**

"Despite how" was changed to "Although" in the revised manuscript on Page 10, Line 413 of revised manuscript.

**L595: what is your interpretation of the finding that the modes found are not harmonics of 12 months? Given that the seasonal (12 month) cycle is important here and many of the other modes may be coupled to it, it would be useful to explain to the reader why other modes emerge as important.**

Although understanding the specific dynamics underlying the nonlinear modes is beyond the scope of paper, the revised manuscript includes references to studies that focused on understanding nonlinear ENSO dynamics (Page 2, Line 76).

**L610: this is an important point which the authors should consider emphasizing in the abstract**

The authors agree that it is important point. As such, the finding will be discussed in the revised abstract.

**Figure 5: consider adding color**

We agree that this figure could be clearer. We thickened lines and changed the line color to red and blue shades designed to be distinguishable by the colorblind.

**Figure 6: consider plotting the global (average) wavelet spectrum adjacent**

Although the authors agree that the global wavelet power spectra would help highlight the dominant peaks in the wavelet power spectra, we could not find any reason to discuss them in the text because we are concerned with changes in wavelet power. For the sake of brevity, we decided to omit them.

**Figure 7: please fix titles**

Thank you for referring us to the errors in the titles. The problem was corrected in the revised manuscript.

**Figure 8: the figure has gotten clipped at the left margin.**

Thank you for referring us to the clipping problem. The problem was corrected in the revised manuscript.

**Figure 9: this is the wrong place to bring this up but it would be helpful to add some discussion in the**
**methods section, specifically around hypothesis testing, about what the 5% cumulative area-wise**
**significance means and how to interpret it.**

The authors agree that some discussion is warranted given the novelty of the statistical tests. A short
discussion was added on Page 4, Line 207 of the revised manuscript, but the reader is referred to Schulte
(2019) for more details.

**Figure 11: please clarify why these pairs were chosen**

As stated on Line 481, the pairs were chosen because they are local maxima in auto-bicoherence that are
statistically significant. Choosing local maxima allows the spatial patterns shown in Figure 11 to emerge
more clearly. A sentence clarifying our choice of pairs was inserted into the revised manuscript on Page
11, Line 466.

**GENERAL COMMENTS The manuscript investigates the relationship between rainfall and ENSO index using wavelet analysis and a wavelet coherence method is proposed to explain the changes of temporal correlation between two time series. The topic of the study is interesting may be outside the scope of the journal. This appears more so as almost all references are from climate journals where this paper sits more naturally. I think this paper should be withdrawn and submitted to an appropriate climate journal, or else reformatted to represent better arguments as to why it is of interest to hydrology directly. I have, however, read through the paper and have some comments that may help the authors publish this successfully.**

Our manuscript addresses precipitation variability in the South Asian monsoon region. As precipitation in this region is critical to mountain glaciers, transboundary rivers, groundwater recharge, and functioning of many ecosystems and human systems, we believe that the topic is of considerable importance to hydrologists and Earth system scientists. However, we appreciate the reviewer's concern, and we recognize that we did not sufficiently emphasize the hydrological relevance of this work in the original manuscript. The first two paragraphs of the introduction section have been rewritten in the revised manuscript to better empathize the hydrological relevance. Many new references have been added as well.

**COMMENT1: Section 2, the way authors computed the monthly anomaly by subtracting the data from the whole period is not the recommended and standard way. It is recommended by WMO that a fixed reference period is defined as the 30-year period 1 January 1961 to 31 December 1990. Authors should consider use this as baseline period, especially when compared SST and Nino indices over different regions.**

Although the authors agree that a 30-year base period is commonly used in climate studies, it is unclear how using the standard base period would benefit the present analysis. Using a different base period would only translate the time series up or down uniformly and would not alter the actual behavior of the time series. Therefore, the results of the wavelet analysis, which are the focus of the present study, would be unchanged. Furthermore, skewness would be made relative to that base period, which would make the more recent skewed events appear less prominent. As such, we feel that subtracting the long-term means to calculate anomalies is the best approach for the present study. By using long-term means, it is easier to see how skewness how evolved throughout the study period, which is consistent with what the wavelet analysis is quantifying.

**COMMENT2: Section 3.1, there is no such Reference Schulte and Lee (2019). More importantly, the reason of adapting Event Decomposition is not well explained and how it helps quantifying the nonlinearity (i.e. skewness) of rainfall and ENSO index is not demonstrated. In the end of Method section, authors considered a synthetic example to illustrate the concept of nonlinear coherence using original time series but following your methodology it should be transformed to event spectra before calculating the coherence. The impact of Event Decomposition on the wavelet analysis and coherency is unknown.**

The authors agree that it is unclear how the event spectra benefit the paper. As such, the event decomposition method results have been removed from the revised manuscript. This removal allowed us to focus more on wavelet analysis, which is the main topic of the paper. The authors note that the coherence spectra are based on the actual time series and not on the event transformed time series.

**COMMENT3: Section 4.2, the relation between skewness and correlation is not explicitly demonstrated. There is a sharp decrease of skewness of June-September rainfall around 1991. Is there any particular reason? And what is the implication of this change? "A comparison of Figures 5a and Figures 5c reveals that the weakening and reversal of the relationship occurs during the time period when the Niño 1+2 index is especially skewed, suggesting that ENSO skewness changes could be contributing to changes in the time-domain correlation between ENSO and All-India rainfall. " This conclusion is in doubt, Figure 5a doesn't include the skewness of August-September rainfall.**

The main reason for showing skewness is because two time series can only be perfectly correlated if all the statistical moments are correlated. More specifically, if the skewness of one time series is increasing but another remains nearly constant, then the lack of correlation between the skewness of the two time series must be contributing to changes in the correlation between the time series. This idea was made more explicit in the revised manuscript (Page 6, Line

229). The sharp decrease in skewness around 1991 could be noise. A full understanding of all sources of Indian rainfall
skewness would require additional analyses, which would digress from the focus of the paper, which is to relate ENSO
skewness to Indian rainfall skewness. Nevertheless, an implication of the result is that noise also influences the
correlation between Indian rainfall and ENSO. This possibility was mentioned in the revised manuscript on Page 9
Line 351.

**COMMENT4: Section 4.2 and 4.3, through the global and local auto-bicoherence analysis, they show the**
**nonlinearity of ENSO indices and India rainfall in the frequency and spatial space individually. But how these**
**two related to each other, authors do not explain explicitly.**

The main reasons for showing the local and global bicoherence analyses is to highlight their differences. Because the
nonlinear coherence between them is weak, we expected that the differences to be large. On page 12, Line 496 of the
revised manuscript, we mentioned how changing ENSO nonlinearity could explain the more frequency occurrence of
Central Pacific El Nino in recent decades, connecting the spatial pattern of auto-bicoherence to local auto-bicoherence.

**Using the nonlinear wavelet coherence method to test your hypothesis should be the major contribution of your**
**work, however it is only briefly discussed in the very end of Section 4.3.2. There are lots of redundant**
**information in the manuscript, which makes the paper long and difficult to read.**

In the revised manuscript, we expanded the nonlinear wavelet coherence method section. In addition, some text was
moved or deleted (e. g Section 4.2.2) of original manuscript) so that the nonlinear coherence section appears earlier
in the revised manuscript. The authors agree that there is a lot of redundant information, which was reduced in the
revised manuscript by deleting text (e. g Section 4.2.2 of original manuscript) and moving some information to the
supplementary material.  For example., Section 4.2.1 of the original manuscript is now the final section of the revised
manuscript. We also now only focus on the All-India time series because looking at different regions of India is not
necessary to get our key message across.

**SPECIFIC POINTS:**

**1. EL Nino or El Nino, please keep it consistent throughout the paper.**

The authors appreciate the identification of this inconsistency, which was corrected throughout the paper. The format
was changed to "El Nino" throughout.

**2. Line 104, keep the numbering format consistent.**

The numbering inconsistency was corrected in the revised manuscript.

**3. Please have a careful look of the format of your references.**

The authors appreciate the comment about the reference formatting. The reference formatting was corrected in the
revised manuscript.

**4. Line 274, keep the equation numbering format consistent.**

The equation formatting inconsistency was corrected in the revised manuscript.

**5: Line 175, Because theory supports a casual link...Authors do not explain this point well. Does strong**
**coherence or association mean causality in nonlinear system? More details are needed.**

Authors agree that more details are needed regarding the causal linkage statement. There are many studies that have
linked ENSO to the Indian monsoon. The Ropelewski and Halpert (1987) study was referenced in the revised
manuscript on Page 4, Line 155 and linkage between ENSO and Indian rainfall through the Walker Circulation is now
mentioned on Page 4, Line 155.

**6. Figure 7, what monsoon rainfall is used, full monsoon or late monsoon?**

The authors note that Figure 7 is the wavelet power spectrum of the All-India time series without any seasonal averaging.

**7. Line 447, what is the abbreviation of AIR standing for?**

The acronym stands for All-India Rainfall. The acronym is now presented when we discuss the All-India rainfall time series in the data section.

**8. Line 135-137, keep the font format consistent.**

The authors appreciate the identification of the formatting issue, which was corrected in the revised manuscript.

**I recommend authors to do a search on [hydrology and wavelets and precipitation and "el Nino"] or maybe "low frequency variability" and see how they have established the link of their paper to the hydrology audience they are presenting to. It may give authors a good idea of how they could improve their pitch.**

The authors appreciate the suggestion regarding a literature search. We conducted a literature search on wavelet analysis and hydrology and included the following references on Page 4 Line 142 because they focus on hydrological applications of wavelet coherence.

[revised manuscript text omitted]

The stronger relationship between All-India rainfall and the Niño 4 index compared to the Niño 1+2 index
relationship with All-India rainfall after the 1970s is more evident in the August-September analysis (Figure 5b). An
abrupt weakening of the Niño 1+2-All-India rainfall relationship occurs around the 1970's, with the relationship
reversing around the 1990s. A comparison of Figures 4b and Figures 5b reveals that the weakening and reversal of
the relationship occurs during the time period when the Niño 1+2 index is especially skewed. The fact that the Niño
1+2 skewness is greater than Niño 4 skewness after 1970s and that the August-September Niño 1+2 index relationship
with August-September All-India rainfall weakens more abruptly than the August-September Niño 4 index
relationship with August-September All-India rainfall suggests that changes in ENSO 
[revised manuscript text omitted]

[Figure]

Figure 3. The (a) time series of the and (b) event spectrum of the (a) Niño 1+2 and (b) index.indices.

[Figure]

Figure 4 Figure 5. 20-year sliding skewness of June-September All-India rainfall and time series for the

Niño 1+2 and Niño 4 indices. (b) 20-year sliding correlation between anomalies for June-September All-

India rainfall and the time series for the Niño 1+2 and Niño 4 indices. (c) Same as (b) but for August-

September All-India rainfall.

[Figure]

Figure 5. 20-year sliding skewness of (a) June-September and (b) August-September All-India rainfall and
time series for the Niño 1+2 and Niño 4 indices.

[Figure]

Figure 6. Wavelet Power spectrum of the (a) Niño 1+2, (b) Niño 3, (c) Niño 3.4, and (d) Niño 4 indices.
Contours enclose regions of 5% cumulative area-wise significance. Light-shaded region represents the
cone of influence, which is the region where edge effects are non-negligible.

[Figure]

Figure 7. Wavelet coherence spectrum between All-India rainfall anomalies and time series for the (a)
Niño 1+2 and (b) Niño 4 indices. Contours enclose regions of 5% cumulative area-wise significance. Light-
shaded region represents the cone of influence, which is the region where edge effects are non-negligible.

[Figure]

Figure 8. (a) The wavelet coherence between All-India rainfall and the Niño 1+ 2 index, the auto-
bicoherence of the Niño 1+2 index, and the nonlinear coherence between the Niño 1+2 index and All-
India rainfall anomalies averaged in the period band of 16 to 64 months. (b) The same as (a) but with the
Niño 4 index.

[Figure]

Figure 9. Global auto-bicoherence spectra of the (a) Niño 1+2, (b) Niño 3, (c) Niño 3.4, and (d) Niño 4
indices. Contours enclose regions of 5% cumulative area-wise significance.

[Figure]

Figure 10. Global auto-bicoherence corresponding to the pairs (a) (31, 31), (b) (56, 56), and (c) (105, 47)
[months].

[Figure]

Figure 11. Global auto-bicoherence corresponding to the pairs (a) (158, 43), (b) (148, 105), (c) (62, 44),
and (d) (88,88) [months].

 Figure 12. Global auto-bicoherence spectra of the (a) All-India, (b) Peninsula, (c) Northwest, (d) Northeast,
(e) West Central, and (f) Central Northeast time series. Contours enclose regions of 5% cumulative area-
wise significance.

Figure 13. Local auto-bicoherence spectra of the (a) Niño 1+2, (b) Niño 3, (c) Niño 3.4, and (d) Niño 4
indices. Contours enclose regions of 5% cumulative area-wise significance and the light shading represents
the cone of influence.

[Figure]

Figure 14. Local auto-bicoherence spectra of the (a) All-India, (b) Peninsula, (c) Northwest, (d) Northeast, (e) West Central, and (f) Central Northeast time series. Contours enclose regions of the 5% cumulative area-wise significance and the light shading represents the cone of influence.

[Figure]

Figure 15. Nonlinear wavelet coherence between the All-India time series and times series for the (a) Niño

1+2, (b) Niño 3, (c) Niño 3.4, and (d) Niño 4 indices. Contours enclose regions of 5% cumulative area-wise significance and light shading represents the cone of influence.

Table 1. Wavelet quantities and what they measure.

|  |  |
| --- | --- |
| Linear Coherene | Measures the correlation between two time series at a particular time scale. |
| Global Auto-bicoherence | Measures the time-averaged |
| Local Auto-bicoherence | Measures the degree of nonlinear |
| Nonlinear Coherence | Measures the cross-correlation between nonlinear modes |

---

## Referee Report (RR1)

**Comments on Manuscript WRR-2020WR027212: A Skewed perspective of the Indian rainfall-ENSO Relationship**

Justin Schulte        Fredrick Policielli        Benjamin Zaitchik

Comments submitted by James Doss-Gollin to HESS
on June 15, 2020

The purpose of this paper is to apply novel methods for bivariate, nonlinear wavelet analysis to understand whether apparent changes in the relationship between indices for ENSO and the Indian Monsoon represent fundamental changes in their relationship. Since the previous submission, the authors have thoughtfully responded to all my comments. I recommend this paper for publication in HESS, pending very minor corrections.

- providing a link to the R software is helpful – I suggest either archiving the particular version of the software used for this code on a repisotry like Zenodo that generates a permanent DOI, and/or including the code for the online supporting information with this article

- the distinction between "linear coherence" and bicoherence is helpful

- wording and formatting comments were addressed and the overall organization of the manuscript was streamlined

- clarification of the types of nonlinearities that the analysis can identify is useful

- figures are improved

- response to questions about the quality of rainfall and ENSO data is reasonable. The authors should note that this analysis neglects uncertainties in the data themselves, but since the objective is to demonstrate the wavelet method, addressing uncertainties in SST reconstructions should not be a priority

- discussion of the choice of Morlet wavelet with $\omega = 6$ is reasonable

The document needs some copy editing – this is beyond the scope of this review although some comments are provided for the abstract:

**L10** "temporally changes" should be "changes in time"

**L11** "changing nonlinear characteristics" is a bit unclear – specify

**L16** "India sub-continent" should be "Indian"

---

## Author Response (AR3)

**Summary**

The authors appreciate the additional comments and suggestions, which have been incorporated in the revised manuscript. The only large change to the manuscript was the removal of the local auto-bicoherence results for the Nino 3 and Nino 3.4 indices. Changes to the manuscript are highlighted in green and deletions are highlighted in red. Page and line numbers correspond to the tracked version of the manuscript. Reviewer responses are in bold text and our response are in plain text.

**Reviewer 1**

**The purpose of this paper is to apply novel methods for bivariate, nonlinear waveletanalysis to understand whether apparent changes in the relationship between indices for ENSO and the Indian Monsoon represent fundamental changes in their relationship. Since the previous submission, the authors have thoughtfully responded to all my comments. I recommend this paper for publication in HESS, pending very minor corrections.**

**• providing a link to the R software is helpful – I suggest either archiving the particular**

**version of the software used for this code on a repisotry like Zenodo that generates a**

**permanent DOI, and/or including the code for the online supporting information with**

**this article**

The authors appreciate the suggestions for the possible places to post the software. The code will be posted online in a repository if the manuscript is accepted.

**• the distinction between "linear coherence" and bicoherence is helpful**

**• wording and formatting comments were addressed and the overall organization of the**

**manuscript was streamlined**

**• clarification of the types of nonlinearities that the analysis can identify is useful**

**• figures are improved**

**• response to questions about the quality of rainfall and ENSO data is reasonable. The**

**authors should note that this analysis neglects uncertainties in the data themselves,**

**but since the objective is to demonstrate the wavelet method, addressing uncertainties**

**in SST reconstructions should not be a priority**

**• discussion of the choice of Morlet wavelet with ω = 6 is reasonable**

**The document needs some copy editing – this is beyond the scope of this review although**

Extensive copy editing was applied to the revised manuscript.

**some comments are provided for the abstract:**

**L10 "temporally changes" should be "changes in time"**

"temporally changes" has been changed to "changes in time" on Line 10.

**L11 "changing nonlinear characteristics" is a bit unclear – specify**

"changing nonlinear characteristics" has been changed to "skewness" on Line 11 to be more specific.

**L16 "India sub-continent" should be "Indian"**

India sub-continent" has been changed to "Indian" on Line 16.

**Reviewer 2**

**GENERAL COMMENTS**

**The manuscript has been significantly improved, but there are some points where it could still be improved for more clarity and readability. My comments are given below and some examples that the authors could consider to make the paper more understandable to readers.**

**COMMENT: In Introduction, the prediction of South Asian Monsoon using ENSO is problematic given their complex relationship in time domain and frequency domain. However, there have been new studies accounting for differences in spectral attributes which improve prediction performance using wavelet analysis. For example,**

**Jiang, Z., Sharma, A., & Johnson, F. (2020). Refining Predictor Spectral Representation Using Wavelet Theory for Improved Natural System Modeling. Water Resources Research, 56(3), e2019WR026962. doi:10.1029/2019wr026962**

**Authors may want to have a look and include some discussion in Introduction as a possible means of improving their assessment further through the latest refinements in wavelet methodology.**

We appreciate the reference to this relevant paper. We have added a brief discussion about the paper on Page 9 Line 65.

**COMMENT1: The way to compute monthly anomaly = monthly – monthly climatology over a baseline period, so given different base period in most cases it won't be uniformly up or down. Authors must have misunderstood something referring to Line 421-423:**

**"To remove the influence of the annual cycle, the time series was converted into anomaly time series by subtracting the 1871-2016 long-term mean for each month from the individual monthly values."**

We appreciate the clarification regarding the chosen base period. However, we still feel it is best to use long-term means because it makes it easier to compare recent El Nino events to others that have occurred in the past.

**COMMENT2: Authors focus on the skewness, but throughout there is not clear definition or quantification of skewness you used in this work. Only a paragraph discussion is given in the introduction (Line 370 to 382). I would suggest adding a description in the Method section.**

The mathematical definition of skewness is now included in the methods section of the revised manuscript on Pages 4-5  Line 175-180.

**COMMENT3: Section 4.2, you probably mess up with figure captions. Figure 4 shows the correlation while Figure 5 gives skewness. Same in Section 4.3. Also, I would suggest authors combine section 4.1 and 4.2 since they both discussed the relationship between the correlation and skewness.**

We appreciate the identification of the errors in the figure captions, which have been corrected in the revised manuscript. The authors agree that Sections 4.1 and 4.2 can be combined. The two sections are now combined and section numbers throughout the manuscript have been adjusted accordingly.

**COMMENT4: Section 4.3, I would suggest authors keep the x-axis labels of Figure 4 and 5 consistent with Figure 6 and 7. So readers can easily refer to each other. Also, the auto-bicoherence of Nino 3 and 3.4 are not shown in the main context or supporting information. However, one entire paragraph is discussed about this. I would suggest authors to add the results of Nino3 and 3.4 to both Figure 7 and 8.**

X labels for all figures have been standardized to help readers refer to plots. For consistency, the results for Nino 3 and Nino 3.4 are not shown to help the reader focus on the results for Nino 1+2 and Nino 4 indices. We decided to make this change because other sections do not show results for the Nino 3 and Nino 3.4 indices. The one exception is for Section 4.5 because the Nino 3.4 index findings in the full auto-bicoherence spectrum are needed to make the spatial auto bicoherence plots. The paragraph discussing the Nino 3 and Nino 3.4 index local auto-bicoherence results was deleted, though some discussion of the results was moved to the discussion section.

**COMMENT5: Section 4.4, the second paragraph is talking about Figure 8 not Figure 10 (There are many other places in the manuscript, e.g. Section 4.5). This might be due to the review**

**process, but authors need to read through the paper and make sure the content and figure are associated with each other.**

Figure references have been corrected throughout the manuscript.

**COMMENT6: Figure 12-15 now are not discussed in the revised manuscript, so it can be removed. Even though Table 1 is included in the end, it is not cited or discussed anywhere in the main context. A discussion should be added to clarify the difference among them.**

The authors are unsure how to address this comment because the revised manuscript contains 13 figures. Table 1 is cited in the methods section (e.g. Line 151 Page 4) and descriptions to the methods are provided in the main text and Table 1.

**SPECIFIC POINTS:**

**1. Line 396: keep the numbering format consistent. 1) -> (1)**

Formatting has been corrected throughout the manuscript.

**2. Where is abbreviation AIR in Data section?**

The abbreviation is located on Page 3 Line 119.

**3. Line 454: 3.1 Wavelet Analysis and hereafter**

It is unclear to the authors what change is suggested.

**4. Line 464: Readers are referred to…**

The word "is" was changed to "are".

**5. Line 588: Eq. (12)**

The equation number was corrected but was adjusted because of the addition of the skewness equation.

**6. Line 517: should be B_local(s1,s1) ?**

Texted has been change to B_local (s1, s1). Note that the notation has been changed to Bn to make equations easier to read.

**7. Line 526: shoulde be phi_n(s1, s1) ?**

Text has been changed to phi_n(s1, s1)

**8. Line 629: small capital w_F(t)**

Notation is now consistent throughout the text. Capitalization was used because that is what is used in the equations.

**9. Figure 3: ylab: Nino 1+2**

Axis label was corrected.

**10. Line 688: it should be Nino 4 time series?**

Because of line number discrepancies between the revised manuscript and those reported in this review, we could not find the suggested change. A careful reading of the manuscript did not identify any such error though.

**11. Line 713: should it be (Figure 4a and 4b)**

Because of line number discrepancies between the revised manuscript and those reported in this review, we could find the suggested change.

**12. Figure 10 and 11: The ylab is from 20N to 20S.**

The axis labels have been corrected.

**13. Line 1117: The R software link missing. These methods…**

The link is provided on Page 12 Line 521 of the revised manuscript.